# Worst-Case Optimal Multi-Armed Gaussian Best Arm Identification with a Fixed Budget

## Abstract

This study investigates the experimental design problem for identifying the arm with the highest expected outcome, referred to as *best arm identification* (BAI). In our experiments, the number of treatment-allocation rounds is fixed. During each round, a decision-maker allocates an arm and observes a corresponding outcome, which follows a Gaussian distribution with variances that can differ among the arms. At the end of the experiment, the decision-maker recommends one of the arms as an estimate of the best arm. To design an experiment, we first discuss lower bounds for the probability of misidentification. Our analysis highlights that the available information on the outcome distribution, such as means (expected outcomes), variances, and the choice of the best arm, significantly influences the lower bounds. Because available information is limited in actual experiments, we develop a lower bound that is valid under the unknown means and the unknown choice of the best arm, which are referred to as the worst-case lower bound. We demonstrate that the worst-case lower bound depends solely on the variances of the outcomes. Then, under the assumption that the variances are known, we propose the *Generalized-Neyman-Allocation (GNA)-empirical-best-arm (EBA) strategy*, an extension of the Neyman allocation proposed by Neyman (1934). We show that the GNA-EBA strategy is asymptotically optimal in the sense that its probability of misidentification aligns with the lower bounds as the sample size increases infinitely and the differences between the expected outcomes of the best and other suboptimal arms converge to the same values across arms. We refer to such strategies as asymptotically worst-case optimal.

## 1 Introduction

Experimental design is crucial in decision-making (Fisher, 1935; Robbins, 1952). This study investigates scenarios involving multiple *arms*[1], such as online advertisements, slot machine arms, diverse therapeutic strategies, and assorted unemployment assistance programs. The objective of an experiment is to identify the arm that yields the highest expected outcome (the *best arm*), while minimizing the probability of misidentification. This problem has been examined in various research areas under a range of names, including *best arm identification* (BAI, Audibert et al., 2010), *ordinal optimization* (Ho et al., 1992)[2], *optimal budget allocation* (Chen et al., 2000), and *policy choice* (Kasy & Sautmann, 2021). We mainly follow the terminologies in BAI. BAI has two formulations: fixed-budget and fixed-confidence BAI; this study focuses on the fixed-budget BAI[3].

---

[1]The term 'arm' is used in the bandit literature (Lattimore & Szepesvári, 2020). There are various names for this concept, including 'treatment arms' (Nair, 2019; Athey & Imbens, 2017), 'policies' (Kasy & Sautmann, 2021), 'treatments' (Hahn et al., 2011), 'designs' (Chen et al., 2000), 'populations' (Glynn & Juneja, 2004), and 'alternatives' (Shin et al., 2018).

[2]Ordinal optimization often considers non-adaptive experiments, and BAI primarily addresses adaptive experiments. However, there are also studies on adaptive experiments in ordinal optimization and non-adaptive experiments in BAI.

[3]The fixed-confidence BAI formulation resembles sequential testing, where a sample size is a random stopping time.

## 2 Problem Setting

We consider a decision-maker who conducts an experiment with a fixed number of rounds $T$, referred to as a sample size or a *budget*, and a fixed set of arms $[K] := 1, 2, \ldots, K$. In each round $t \in [T] := 1, 2, \ldots, T$, the decision-maker allocates arm $A_t \in [K]$ to an experimental unit and immediately observes outcome $Y_t$ linked to the allocated arm $A_t$. The decision-maker's goal is to identify the arm with the highest expected outcome, minimizing the probability of misidentification at the end of the experiment.

**Potential outcomes.** To describe the data-generating process, we introduce potential outcomes following the Neyman-Rubin model (Neyman, 1923; Rubin, 1974). Let $P$ be a joint distribution of $K$-potential outcomes $(Y^1, Y^2, \ldots, Y^K)$. Let $\mathbb{P}_P$ and $\mathbb{E}_P$ be the probability and expectation under $P$, respectively, and let $\mu^a(P) = \mathbb{E}_P[Y^a]$ be the expected outcome. Let $a^*(P) := \arg\max_{a \in [K]} \mu^a(P)$ be the best arm under $P$, and $\mathcal{P}$ be the set of all possible joint distributions such that $a^*(P)$ is unique for all $P \in \mathcal{P}$. This study focuses on an instance $P \in \mathcal{P}$ where $(Y^1, Y^2, \ldots, Y^K)$ follows a multivariate Gaussian distribution with a unique best arm $a^*(P) \in [K]$, defined as

$$\mathcal{P}^{\mathrm{G}}(a^*, \underline{\Delta}, \overline{\Delta}) := \left\{ P \in \mathcal{P} \mid \forall a \in [K] \ \ Y^a \sim \mathcal{N}\left(\mu^a, (\sigma^a)^2\right), \right.$$

$$\left. \forall a \in [K] \ \ \mu^a \in \mathbb{R}^K, \ \ \forall a \in [K] \ \ \mu^a \ \ (\sigma^a)^2 \in [\underline{C}, \overline{C}]^K, \ \ \forall a \in [K]\backslash\{a^*\} \ \underline{\Delta} \leq \mu^{a^*} - \mu^a \leq \overline{\Delta} \right\},$$

where $\mathcal{N}(\mu, v)$ is a Gaussian distribution with a mean (expected outcome) $\mu \in \mathbb{R}$ and a variance $v > 0$, $\underline{\Delta}$ and $\overline{\Delta}$ are constants independent of $T$ such that they are lower and upper bounds for a gap $\mu^{a^*} - \max_{a \in [K]\backslash\{a^*\}} \mu^a$ ($0 < \underline{\Delta} \leq \overline{\Delta} < \infty$ holds), and $\underline{C}$ and $\overline{C}$ are *unknown* universal constants such that $0 < \underline{C} < \overline{C} < \infty$. Note that $\underline{C}$ and $\overline{C}$ are just introduced for a technical purpose to assume that $(\sigma^a)^2$ is bounded, and we do not use them in designing algorithms. We refer to $\mathcal{P}^{\mathrm{G}}(a^*, \underline{\Delta}, \overline{\Delta})$ as a *Gaussian bandit model* [4]

**Experiment.** Let $P_0 \in \mathcal{P}^{\mathrm{G}}(a^*, \underline{\Delta}, \overline{\Delta})$ be an instance of bandit models that generates potential outcomes in an experiment, which is decided in advance of the experiment, and fixed throughout the experiment. The decision-maker knows the true values of the variances $((\sigma^a)^2)_{a \in [K]}$ but does not know $a^*$ and $(\mu^a)_{a \in [K]}$. We use $P_0$ when emphasizing the dependency on the data-generating process (DGP). An outcome in round $t \in [T]$ is $Y_t = \sum_{a \in [K]} \mathbb{1}[A_t = a]Y_t^a$, where $Y_t^a \in \mathbb{R}$ is a potential independent outcome (random variable), and $(Y_t^1, Y_t^2, \ldots, Y_t^K)$ be an independent (i.i.d.) copy of $(Y^1, Y^2, \ldots, Y^K)$ at round $t \in [T]$ under $P_0$. Then, we consider an experiment with the following procedure of a decision-maker at each round $t \in [T]$:

1. A potential outcome $(Y_t^1, Y_t^2, \ldots, Y_t^K)$ is drawn from $P_0$.
2. The decision-maker allocates an arm $A_t \in \mathcal{A}$ based on past observations $\{(Y_s, A_s)\}_{s=1}^{t-1}$.
3. The decision-maker observes a corresponding outcome $Y_t = \sum_{a \in \mathcal{A}} \mathbb{1}[A_t = a]Y_t^a$

At the end of the experiment, the decision-maker estimates $a^*(P_0)$, denoted by $\widehat{a}_T \in [K]$. Here, an outcome in round $t \in [T]$ is $Y_t = \sum_{a \in [K]} \mathbb{1}[A_t = a]Y_t^a$.

**Probability of misidentification.** Our goal is to minimize the *probability of misidentification*, defined as

$$\mathbb{P}_P(\widehat{a}_T \neq a^*(P)).$$

To evaluate the exponential convergence of $\mathbb{P}_P(\widehat{a}_T \neq a^*(P))$ for any $P$, we employ the following measure, called the *complexity*:

$$-\frac{1}{T} \log \mathbb{P}_P(\widehat{a}_T \neq a^*(P)).$$

---

[4]Kaufmann et al. (2016) considers a Gaussian bandit model $\mathcal{P} := \left\{ \mathcal{P}^{\mathrm{G}}(a^*, \underline{\Delta}, \overline{\Delta}) \subset \mathcal{P} \mid \forall a^* \in [K] \right\}$. In this study, we explicitly define the existence of universal constants $\underline{C}$, $\overline{C}$, $\underline{\Delta}$, and $\overline{\Delta}$. Although Kaufmann et al. (2016) omits them, their results do not hold without such constants; that is, boundedness of the means and variances are essentially required. For example, if variances are zero or infinity, their lower bounds become infinity or zero. Additionally, under such cases, their proposed strategy ($\alpha$-Elimination) becomes ill-defined, since their allocation rule cannot be defined well.

The complexity $-\frac{1}{T}\log\mathbb{P}_P(\widehat{a}_T \neq a^*)$ has beeen widely employed in the literature on ordinal optimization and BAI (Glynn & Juneja, 2004; Kaufmann et al., 2016). In hypothesis testing, Bahadur (1960) suggests the use of a similar measure to assess performances of methods in hypothesis testing. Also see Section A.3.

**Strategy.** We define a *strategy* of a decision-maker as a pair of $((A_t)_{t\in[K]}, \widehat{a}_T)$, where $(A_t)_{t\in[K]}$ is the allocation rule, and $\widehat{a}_T$ is the recommendation rule. Formally, with the sigma-algebras $\mathcal{F}_t = \sigma(A_1, Y_1, \ldots, A_t, Y_t)$, a strategy is a pair $((A_t)_{t\in[T]}, \widehat{a}_T)$, where

- $(A_t)_{t\in[T]}$ is an allocation rule, which is $\mathcal{F}_{t-1}$-measurable and allocates an arm $A_t \in [K]$ in each round $t$ using observations up to round $t-1$.

- $\widehat{a}_T$ is a recommendation rule, which is an $\mathcal{F}_T$-measurable estimator of the best arm $a^*$ using observations up to round $T$.

We denote a strategy by $\pi$. We also denote $A_t$ and $\widehat{a}_T$ by $A_t^\pi$ and $\widehat{a}_T^\pi$ when we emphasize that $A_t$ and $\widehat{a}_T$ depend on $\pi$. Let $\Pi$ be the set of all possible strategies.

This definition of strategies allows us to design adaptive experiments where we can decide $A_t$ using past observations. In this study, although we develop lower bounds that work for both adaptive and non-adaptive experiments,[5] our proposed strategy is non-adaptive; that is, $A_t$ is decided without using observations obtained in an experiment. As we show later, our lower bounds depend only on variances of potential outcomes. By assuming that the variances are known, we design a non-adaptive strategy that is asymptotically optimal in the sense that its probability of misidentification aligns with the lower bounds. If the variances are unknown, we may consider estimating them during an experiment and using $\mathcal{F}_{t-1}$-measurable variance estimators at each round $t$. However, it is unknown whether an optimal strategy exists when we estimate variances during an experiment. We leave it as an open issue (Section 8).

**Notation.** When emphasizing the dependency of the best arm on $P$, we denote it by $a^*(P) := \arg\max_{a\in[K]} \mu^a(P)$. Let $\Delta^a(P) := \mu^{a^*(P)}(P) - \mu^a(P)$. For $P \in \mathcal{P}$, let $P^a$ be a distribution of a reward of arm $a \in [K]$.

# 3 Open questions about Optimal Strategies in Fixed-Budget BAI

The existence of optimal strategies in fixed-budget BAI has been a longstanding issue, which is related to tight lower bounds and corresponding optimal strategies.

## 3.1 Conjectures about Information-theoretic Lower Bounds

To discuss tight lower bounds, we consider restricting a class of strategies. Following the existing studies such as Kaufmann et al. (2016), we focus on consistent strategies that recommend the true best arm with probability one as $T \to \infty$.

**Definition 3.1** (Consistent strategy). We say that a strategy $\pi$ is consistent if $\mathbb{P}_{P_0}(\widehat{a}_T^\pi = a^*(P_0)) \to 1$ as $T \to \infty$ for any DGP $P_0 \in \mathcal{P}$. We denote the class of all possible consistent strategies by $\Pi^{\mathrm{cons}}$.

Kaufmann et al. (2016) discusses lower bounds for both fixed-budget and fixed-confidence BAI and presents lower bounds for two-armed Gaussian bandits in fixed-budget BAI. Based on their arguments, Garivier & Kaufmann (2016) presents a lower bound and optimal strategy for fixed-confidence BAI. They also conjecture a lower bound for fixed-budget BAI as $-\frac{1}{T}\log\mathbb{P}_P(\widehat{a}_T \neq a^*(P)) \leq \max_{w\in\mathcal{W}} \Gamma^*(P, w)$ for all

---

[5]Non-adaptive experiments are also referred to as static experiments. The difference between adaptive and non-adaptive experiments is the dependency on the past observations. In non-adaptive experiments, we first fix $\{A_t\}_{t\in[T]}$ at the beginning of an experiment and do not change it. Both in adaptive and non-adaptive experiments, $\widehat{a}_T$ depends on observations $\{(A_t, Y_t)\}_{t=1}^T$.

Table 1: Comparison of lower and upper bounds. Among several metrics for fixed-budget BAI, we compare our proposed the worst-case metric $\min_{a^* \in [K]} \inf_{P \in \mathcal{P}^{\mathrm{G}}(a^*, \underline{\Delta}, \overline{\Delta})} \limsup_{T \to \infty} -\frac{1}{T} \log \mathbb{P}_P(\widehat{a}_T^\pi \neq a^*)$ with $-\frac{1}{T} \log \mathbb{P}_P(\widehat{a}_T \neq a^*)$, $\inf_{P \in \mathcal{P}} H(P) \liminf_{T \to \infty} -\frac{1}{T} \log \mathbb{P}_P(\widehat{a}_T \neq a^*)$ (minimax), and (Bayes). In the table, the "Bounded models" denotes the set of distributions with bounded mean parameters, and the "Bernoulli models" denotes the set of Bernoulli distributions. The "General models" denotes a set of general distributions, and we omit the detailed definitions (see each study). In all cases, we consider bandit models in which only mean parameters vary, including Gaussian models with fixed variances. We use the following quantities: $\Gamma^*(P, w) \coloneqq \min_{a \in [K] \setminus \{a^*(P)\}} \inf_{\mu^a < v^a < \mu^{a^*}(P)} \left\{ w(a^*(P)) \widetilde{\mathrm{KL}}\big(v^a, \mu^{a^*(P)}(P)\big) + w(a) \widetilde{\mathrm{KL}}\big(v^a, \mu^a(P)\big) \right\}$ and $\Lambda^*(P, w, H) \coloneqq \inf_{\mu^a < v^a < \mu^{a^*}(P)} H(P) \sum_{a \in [K]} w(a) \widetilde{\mathrm{KL}}\big(v^a, \mu^a(P)\big)$, where $\widetilde{\mathrm{KL}}$ denotes the KL divergence between two distributions whose means are $(\mu)_{a \in [K]}$ and $(\mu^a(P))_{a \in [K]}$ and $H(P)$ denotes some quantity that represents the difficulty of the problems. The definition of $H(P)$ differs among existing studies.

| | Lower and upper Bounds | | Multi-arm ($K \geq 3$) | Strategy class | Bandit model $\mathcal{P}$ |
|---|---|---|---|---|---|
| | Lower bound (Upper bound of ) | Upper bound (Lower bound of ) | | | |
| **Metric** | $-\frac{1}{T} \log \mathbb{P}_P(\widehat{a}_T \neq a^*(P))$. | | | | |
| Glynn & Juneja (2004) | - | $\max_{w \in \mathcal{W}} \Gamma^*(P_0, w)$ | ✓ | - | General models (Full information) |
| Kaufmann et al. (2016) | $\frac{(\mu^1 - \mu^2)^2}{2(\sigma^1 + \sigma^2)^2}$ | $\frac{(\mu^1 - \mu^2)^2}{2(\sigma^1 + \sigma^2)^2}$ | $K = 2$ | Cons. | General models |
| Garivier & Kaufmann (2016) (Conjecture) | $\max_{w \in \mathcal{W}} \Gamma^*(P, w)$ | - | N/A | N/A | - |
| Carpentier & Locatelli (2016) (Non-asymptotic) | $C_{\mathrm{Carpentier}}\left(\frac{1}{\log(K)H}\right)$ $C_{\mathrm{Carpentier}} \neq D_{\mathrm{Carpentier}}$ are | $D_{\mathrm{Carpentier}}\left(\frac{1}{\log(K)H_2}\right)$ universal constants. | ✓ | Any | Bounded models |
| Ariu et al. (2021) | $\exists P \in \mathcal{P},\ \frac{C}{\log(K)} \max_{w \in \mathcal{W}} \Gamma^*(P, w)$ | - | ✓ | Any | Bernoulli bandits |
| Degenne (2023) (Theorem 1) | $\max_{w \in \mathcal{W}}, \Gamma(P_0, w)$ | $\max_{w \in \mathcal{W}} \Gamma^*(P_0, w)$ (under the full information) | ✓ | Inv. (& Cons.) | General models |
| Wang et al. (2023) | $\forall P \in \mathcal{P}, \Gamma^*(P, w^{\mathrm{Uni}})$ $w^{\mathrm{Uni}} = (1/2, 1/2)$ | $\Gamma^*(P, w^{\mathrm{Uni}})$ | $K = 2$ | Cons. & Stable | Bernoulli |
| **Metric** | $\inf_{P \in \mathcal{P}} H(P) \liminf_{T \to \infty} -\frac{1}{T} \log \mathbb{P}_P(\widehat{a}_T \neq a^*(P))$. | | | | |
| Komiyama et al. (2021) | $\sup_{w \in \mathcal{W}} \inf_{P \in \mathcal{P}} \Lambda^*(P, w, H)$ | Theorem 5 in Komiyama et al. (2022) | ✓ | Any | Bounded |
| Degenne (2023) (Theorem 3) | $\inf_{P \in \mathcal{P}} \max_{w \in \mathcal{W}} \Lambda^*(P, w, H)$ | Theorem 5 in Degenne (2023) | ✓ | Cons. | General models |
| | $\frac{3}{80} \log(K)$ | - | ✓ | Any | Gaussian models with $\sigma^a = 1$ for all $a \in [K]$ |
| **Metric** | $\inf_{P \in \mathcal{P}} \limsup_{T \to \infty} -\frac{1}{T} \log \mathbb{P}_P(\widehat{a}_T^\pi \neq a^*(P))$. | | | | |
| Ours | $\max_{w \in \mathcal{W}} \min_{a^* \in [K]} \frac{\overline{\Delta}^2}{2\Omega^{a^*, a}(w)}$ | $\max_{w \in \mathcal{W}} \min_{a^* \in [K]} \frac{\underline{\Delta}^2}{2\Omega^{a^*, a}(w)}$ The lower and upper bound matches as $\overline{\Delta} - \underline{\Delta} \to 0$. | ✓ | Cons. | Gaussian models with known variances |

$P \in \widetilde{\mathcal{P}}$ for some well-defined bandit models $\mathcal{P}$,[6][7] where

$$\Gamma^*(P, w) \coloneqq \min_{a \in [K] \setminus \{a^*(P)\}} \inf_{\mu^a < v^a < \mu^{a^*(P)}} \left\{ w(a^*(P)) \widetilde{\mathrm{KL}}\big(v^a, \mu^{a^*(P)}(P)\big) + w(a) \widetilde{\mathrm{KL}}\big(v^a, \mu^a(P)\big) \right\}, \quad (1)$$

where $\widetilde{\mathrm{KL}}$ denotes the KL divergence between two distributions parameterized by mean parameters, and $\mathcal{W}$ is the probablity simplex defined as

$$\mathcal{W} = \left\{ w : [K] \to (0, 1) \mid \sum_{a \in [K]} w(a) = 1 \right\}.$$

This conjectured lower bound is a straightforward extension of the basic tool developed by Kaufmann et al. (2016) and an analogy of the lower bound for fixed-confidence BAI shown by Garivier & Kaufmann (2016). However, while a lower bound for two-armed Gaussian bandits is derived for any consistent strategies, this lower bound cannot be derived only under the restriction of any consistent strategies. Summarizing these early discussions, Kaufmann (2020) clarifies the problem and points out that there exists the reverse KL divergence problem, which makes the derivation of the lower bound difficult.

---

[6] In a more rigorous analysis, we should use $\sup_{w \in \mathcal{W}} \Gamma^*(P, w)$ and then discuss the existence of the maximum. However, for simplicity, we use $\max_{w \in \mathcal{W}} \Gamma^*(P, w)$ in literature review.

[7] Note that lower bounds (resp. upper bounds) for $\mathbb{P}_P(\widehat{a}_T^\pi \neq a^*)$ corresponds to upper bounds (resp. lower bounds) for $-\frac{1}{T} \log \mathbb{P}_P(\widehat{a}_T^\pi \neq a^*)$.

## 3.2 Impossibility Theorem

Following these studies, the existence of optimal strategies is discussed by Kasy & Sautmann (2021) and Ariu et al. (2021). Ariu et al. (2021) proves that there exists a distribution $P$ under which a lower bound is larger than the conjectured lower bound $\max_{w \in \mathcal{W}} \Gamma^*(P, w)$ by showing that for any strategies,

$$\exists P \in \mathcal{P}, \; -\frac{1}{T} \log \mathbb{P}_P(\widehat{a}_T \neq a^*(P)) \leq \frac{800}{\log(K)} \sup_{w \in \mathcal{W}} \Gamma^*(P, w),$$

where $\frac{800}{\log(K)} \sup_{w \in \mathcal{W}} \Gamma^*(P, w) \leq \sup_{w \in \mathcal{W}} \Gamma^*(P, w)$ holds when $K$ is sufficiently large. Qin (2022) summarizes these arguments as an open question.

## 3.3 Worst-case Optimal Strategies with the Problem Difficulty

Komiyama et al. (2022) tackles this problem by considering the minimax evaluation of $\limsup_{T \to \infty} -\frac{1}{T} \log \mathbb{P}_P(\widehat{a}_T^\pi \neq a^*(P))$, defined as

$$\inf_{P \in \mathcal{P}} H(P) \liminf_{T \to \infty} -\frac{1}{T} \log \mathbb{P}_P(\widehat{a}_T \neq a^*),$$

where $H(P)$ represents the difficulty of the problem $P$. See Komiyama et al. (2022) for a more detailed definition. This approach allows us to avoid the reverse KL problem in the lower bound derivation.

Degenne (2023) explores the open question and shows that for any consistent strategies, a lower bound is given as $\inf_{P \in \mathcal{P}} H(P) \liminf_{T \to \infty} -\frac{1}{T} \log \mathbb{P}_P(\widehat{a}_T \neq a^*) \leq \max_{w \in \mathcal{W}} \Lambda^*(P, w, H)$ for any $P \in \widetilde{\mathcal{P}}$, where

$$\Lambda^*(P, w, H) \coloneqq \inf_{\mu^a < v^a < \mu^{a^*}(P)} H(P) \sum_{a \in [K]} w(a) \widetilde{\mathrm{KL}}(v^a, \mu^a(P)).$$

Note that Degenne (2023)'s lower bound is tighter than Komiyama et al. (2022)'s.

## 3.4 Asymptotically Invariant (Static) Strategies

A candidate of an optimal strategy is the one proposed by Glynn & Juneja (2004), which is feasible only when we know the full information about the DPG $P_0$. Although their upper bound aligns with the conjectured lower bound $\max_{w \in \mathcal{W}} \Gamma^*(P, w)$, it does not match the lower bounds by Ariu et al. (2021) and Degenne (2023) for any consistent strategies. One of the reason for this difference is the knowledge about the DGP $P_0$; if we do not have the full information and need to estimate it, the upper bound does not match $\max_{w \in \mathcal{W}} \Gamma^*(P, w)$ for the estimation error about the information.

To fill this gap between the lower and upper bounds, we and Degenne (2023) consider restricting the strategy class to asymptotically invariant (static) strategies, in addition to consistent strategies. For an instance $P \in \mathcal{P}$ and a strategy $\pi \in \Pi$, let us define an average sample allocation ratio $\kappa_{T,P}^\pi : [K] \to (0, 1)$ as $\kappa_{T,P}^\pi(a) \coloneqq \mathbb{E}_P\left[\frac{1}{T} \sum_{t=1}^T \mathbb{1}[A_t^\pi = a]\right]$, which satisfies $\sum_{a \in [K]} \kappa_{T,P}^\pi(a) = 1$. This quantity represents the average sample allocation to each arm $a$ over a distribution $P \in \mathcal{P}^{\mathrm{G}}(a^*, \underline{\Delta}, \overline{\Delta})$ under a strategy $\pi$. Then, we define asymptotically invariant strategies as follows.[8]

**Definition 3.2** (Asymptotically invariant strategy). A strategy $\pi$ is called asymptotically invariant if there exists $w^\pi \in \mathcal{W}$ such that for any DGP $P_0 \in \mathcal{P}$, and all $a \in [K]$,

$$\kappa_{T,P_0}^\pi(a) = w^\pi(a) + o(1) \tag{2}$$

holds as $T \to \infty$. We denote the class of all possible consistent strategies by $\Pi^{\mathrm{inv}}$.

---

[8]Note that Degenne (2023) independently considers a similar class of strategies and refers to them as static strategies. Degenne (2023) considers strategies that are completely independent of the DGP $P_0$, while we consider strategies that are asymptotically independent of $P_0$. This is because we aim to include strategies that estimate the variance from $P_0$ under a bandit model with fixed variances. Therefore, we keep our terminology.

Asymptotically invariant strategies allocate arms with the proportion $w^\pi$ under any distribution $P \in \mathcal{P}^{\mathrm{G}}(a^*, \underline{\Delta}, \overline{\Delta})$ as $T$ approaches infinity (see Definition 3.2); that is, the strategies does not depend on the DGP $P_0$ asymptotically.

Degenne (2023) shows that any consistent and asymptotically invariant strategies $\pi$ satisfy

$$\forall P \in \widetilde{\mathcal{P}}, \ -\frac{1}{T} \log \mathbb{P}_P(\widehat{a}_T^\pi \neq a^*) \leq \max_{w \in \mathcal{W}}, \Gamma(P, w),$$

where $\widetilde{\mathcal{P}} \subset \mathcal{P}$ is a well-defined general bandit models, while any strategies $\pi$ satisfy

$$\exists P \in \mathcal{P}_{\sigma=1}^{\mathrm{G}}, \ -\frac{1}{T} \log \mathbb{P}_P(\widehat{a}_T^\pi \neq a^*) \leq \frac{80}{3 \log(K)} \max_{w \in \mathcal{W}}, \Gamma(P, w),$$

where $\mathcal{P}_{\sigma=1}^{\mathrm{G}} := \left\{ \mathcal{P}^{\mathrm{G}}(a^*, \underline{\Delta}, \overline{\Delta}) \subset \mathcal{P} \mid \forall a^* \in [K], \ \forall a \in [K] \ \sigma^a = 1 \right\}.$

Note that although the lower bound is given as $\max_{w \in \mathcal{W}}, \Gamma(P, w)$, the allocation should be constructed to be independent of $P_0$ due to the definition of asymptotically invariant strategies. We can still define the allocation as $w^* = \arg\max_{w \in \mathcal{W}}, \Gamma(P_0, w)$ for the DGP instance $P_0$. However, we can construct $w^*$, depending on $P_0$. In this sense, the meaning of a strategy implied from the lower bound of Degenne (2023) will be significantly different from the strategy by Glynn & Juneja (2004), although they look similar.

This result implies that for any strategies $\pi$ and the problem difficulty defined as $H(P) = \left( \max_{w \in \mathcal{W}}, \Gamma(P, w) \right)^{-1}$,

$$\inf_{P \in \mathcal{P}} H(P) \liminf_{T \to \infty} -\frac{1}{T} \log \mathbb{P}_P(\widehat{a}_T^\pi \neq a^*) < 1,$$

holds; that is, $\max_{w \in \mathcal{W}}, \Gamma(P, w)$ is unattainable unless we assume the access to the full information about the DGP $P_0$.

### 3.5 Open Questions

We list the open questions raised in Kaufmann et al. (2016), Garivier & Kaufmann (2016), Kaufmann (2020), Kasy & Sautmann (2021), Ariu et al. (2021), Qin (2022), Degenne (2023), and Wang et al. (2023). We also summarize related arguments below to clarify what has been elucidated and what remains unelucidated.[9]

**Open question 1.** *Does there exist a desirable strategy class $\widetilde{\Pi}$ and a well-defined function $\widetilde{\Gamma}^* : \mathcal{P} \to \mathbb{R}$ that satisfy the following?: any strategy $\pi \in \widetilde{\Pi}$ satisfies*

$$\forall P \in \mathcal{P}, \quad \liminf_{T \to \infty} -\frac{1}{T} \log \mathbb{P}_P(\widehat{a}_T^\pi \neq a^*(P)) \leq \widetilde{\Gamma}(P),$$

*and there is a strategy $\pi \in \widetilde{\Pi}$ whose probability of misidentification satisfies*

$$\forall P \in \mathcal{P}, \quad \liminf_{T \to \infty} -\frac{1}{T} \log \mathbb{P}_P(\widehat{a}_T^\pi \neq a^*(P)) \geq \widetilde{\Gamma}(P).$$

There are several impossibility theorems for this open question, as follows:

- Ariu et al. (2021) shows that for any strategies $\pi \in \Pi$, $\exists P \in \mathcal{P}$, $-\frac{1}{T} \log \mathbb{P}_P(\widehat{a}_T \neq a^*) \leq \frac{800}{\log(K)} \sup_{w \in \mathcal{W}} \Gamma^*(P, w)$.

- Degenne (2023) shows that for any strategies $\pi \in \Pi$, $\exists P \in \mathcal{P}_{\sigma=1}^{\mathrm{G}}$, $-\frac{1}{T} \log \mathbb{P}_P(\widehat{a}_T \neq a^*) \leq \frac{80}{3 \log(K)} \max_{w \in \mathcal{W}}, \Gamma(P, w)$. Note that $\max_{w \in \mathcal{W}}, \Gamma(P, w) \leq \Gamma^*(P, w)$ holds when $K$ is sufficiently large, and Degenne (2023)'s lower bound is tighter than Ariu et al. (2021)'s lower bound.

---

[9] Degenne (2023) also raises another open question in his Section 4, which focuses on Bernoulli bandits. Because this study focuses on Gaussian bandits, we omit introducing the open question.

Here, $\Gamma^*(P, w)$ corresponds to an upper bound of the strategy proposed by Glynn & Juneja (2004) and a lower bound of consistent strategies shown by Kaufmann et al. (2016). Note that Glynn & Juneja (2004) requires full information about bandit models, including the mean parameters. They show that there is an instance $P \in \mathcal{P}$ such that its lower bound is larger than $\max_{w \in \mathcal{W}} \Gamma^*(P, w)$.

**Open question 2.** *Does there exist a strategy other than the uniform allocation that performs no worse than the uniform allocation for all $P \in \mathcal{P}$?*

A next open question is whether there is a strategy whose probability of misidentification is smaller than those using uniform allocation (allocating arms with equal sample sizes; that is, $\kappa_{T,P}^\pi = 1/K$ for all $a \in [K]$ and any $P \in \mathcal{P}$). For consistent strategies, Kaufmann et al. (2016) notes that a strategy using the uniform allocation is nearly optimal when outcomes follow two-armed Bernoulli bandits. Furthermore, they show that a strategy allocating arms in proportion to the standard deviations of outcomes is asymptotically optimal in two-armed Gaussian bandits if the standard deviations are known. Such a strategy, known as the *Neyman allocation*, is conjectured or shown to be optimal in various senses under Gaussian bandits (Neyman, 1934; Hahn et al., 2011; Glynn & Juneja, 2004; Kaufmann et al., 2016; Adusumilli, 2022).

**Open question 3.** *Does there exist a strategy that strictly outperform the asymptotically invariant strategy?*

Lastly, we introduce an open question about the asymptotically invariant strategy. The open question is whether a strategy that outperforms the asymptotically invariant strategy exists. This problem is raised by existing studies such as Degenne (2023) and Wang et al. (2023). From its definition, the asymptotically invariant strategy cannot depend on $P$. Therefore, as we point out in Sections 5.2 and 5.3, such a strategy would be the uniform allocation or depend on some parameters invariant across $P$. Specifically, in Sections 5.2 and 5.3, we find that the best arm itself should be known in advance of an experiment. Degenne (2023) uses such a strategy as a baseline and compares feasible strategies to it. As a result, Degenne (2023) shows that there exists $P$ under which the probability of misidentification of any strategies cannot align with the lower bound.

## 4 Main Results: Worst-Case Optimal Fixed-Budget BAI with Matching Constants

This section provides our main results. In fixed-budget BAI, the following problems have not been fully explored:

- Ariu et al. (2021) and Degenne (2023) show that there exists $P$ such that a lower bound is larger than $\max_{w \in \mathcal{W}}, \Gamma(P, w)$. However, there is still possibility that for well-chosen $\mathcal{P}$, for any consistent strategy, there exist matching lower and upper bounds related to $\max_{w \in \mathcal{W}} \Gamma^*(P, w)$ (Open questions 1).

- Degenne (2023) discusses the problem difficulty by using a lower bound for static (asymptotically invariant) strategies. However, it is unclear what strategy we obtain from the lower bound $\max_{w \in \mathcal{W}}, \Gamma(P, w)$, which seems to be the same as Glynn & Juneja (2004) but cannot depend on the DGP $P_0$. Is it different from the uniform allocation strategy or the Neyman allocation (Open question 2)?

- Does there exist a strategy that returns a smaller probability of misidentification compared to asymptotically invariant strategies with distributional information (Open questions 2 and 3).

This study tackles these problems by providing the following results:

1. A tight worst-case lower bound for multi-armed Gaussian bandits under restricted bandit models.

2. An asymptotically optimal strategy whose upper bound aligns with the lower bound as the budget approaches infinity and the difference between $\underline{\Delta}$ and $\overline{\Delta}$ approaches zero.

In this section, we first show the lower bound in Section 4.1. Then, based on the lower bound, we develop a non-adaptive strategy in

---

**Algorithm 1** GNA-EBA strategy

---

**Parameter:** Fixed budget $T$.

**Allocation rule: generalized Neyman allocation.**

Allocate $A_t = 1$ if $t \leq \lceil w^{\mathrm{GNA}}(1)T \rceil$ and $A_t = a$ if $\left\lceil \sum_{b=1}^{a-1} w^{\mathrm{GNA}}(b)T \right\rceil < t \leq \left\lceil \sum_{b=1}^{a} w^{\mathrm{GNA}}(b)T \right\rceil$ for $a \in [K] \backslash \{1\}$.

**Recommendation rule: empirical best arm.**

Recommend $\widehat{a}_T^{\mathrm{EBA}}$ following equation 3.

---

## 4.1 Worst-case Lower Bound

In Section 5, we derive lower bounds for strategies based on available information. Specifically, we focus on how the available information affects lower bounds and the existence of strategies whose probability of misidentification aligns with these lower bounds. We find that lower bounds depend significantly on the amount of information available regarding the distribution of rewards for arms prior to the experiment.

From the information theory, we can relate the lower bounds to the Kullback–Leibler (KL) divergence $\mathrm{KL}(Q^a, P_0^a)$ between the DGP $P_0 \in \mathcal{P}$ and an alternative hypothesis $Q \in \cup_{b \neq a^*} \mathcal{P}(b, \underline{\Delta}, \overline{\Delta})$ (Lai & Robbins, 1985; Kaufmann et al., 2016). From the lower bounds, we can compute an ideal expected number of times arms are allocated to each experimental unit; that is, $\kappa_{T,P_0}^\pi$. When the lower bounds are linked to the KL divergence, the corresponding ideal sample allocation rule also depends on the KL divergence (Glynn & Juneja, 2004).

If we know the distributions of arms' outcomes completely, we can compute the KL divergence, which allows us to design a strategy whose probability of misidentification matches the lower bounds of Kaufmann et al. (2016) as $T \to \infty$ (Glynn & Juneja, 2004; Chen et al., 2000; Gärtner, 1977; Ellis, 1984). However, the full information requires us to know which arm is the best arm, and without the knowledge, the strategy is infeasible. Since optimal strategies are characterized by distributional information, the lack of full information hinders us from designing asymptotically optimal strategies. Therefore, we reflect the limitation by considering the worst cases regarding the mean parameters and the choice of the best arm. Specifically, we consider the worst-case lower bound defined as

$$\inf_{P \in \cup_{a^* \in [K]} \mathcal{P}^{\mathrm{G}}(a^*, \underline{\Delta}, \overline{\Delta})} \limsup_{T \to \infty} -\frac{1}{T} \log \mathbb{P}_P(\widehat{a}_T^\pi \neq a^*(P)).$$

While the lower bounds with full information are characterized by the KL divergence (Lai & Robbins, 1985; Kaufmann et al., 2016), the worst-case lower bounds are characterized by the variances of potential outcomes. Hence, knowledge of at least the variances is sufficient to design worst-case optimal strategies.

Then, the following theorem provides a worst-case lower bound. The proof is shown in Appendix D.

**Theorem 4.1** (Best-arm-worst-case lower bound). *For any $0 < \underline{\Delta} \leq \overline{\Delta} < \infty$, any consistent (Definition 3.1) strategy $\pi \in \Pi^{\mathrm{cons}} \cap \Pi^{\mathrm{inv}}$ satisfies*

$$\inf_{P \in \cup_{a^* \in [K]} \mathcal{P}^{\mathrm{G}}(a^*, \underline{\Delta}, \overline{\Delta})} \limsup_{T \to \infty} -\frac{1}{T} \log \mathbb{P}_P(\widehat{a}_T^\pi \neq a^*) \leq \mathrm{LowerBound}(\overline{\Delta}) := \max_{w \in \mathcal{W}} \min_{a^* \in [K]} \frac{\overline{\Delta}^2}{2\Omega^{a^*, a}(w)},$$

*where $\Omega^{a^*, a}(w) = \frac{\left(\sigma^{a^*}\right)^2}{w(a^*)} + \frac{\left(\sigma^a\right)^2}{w(a)}$.*

Additionally, in Section 5.6, we discuss that a lower bound for any consistent strategies is the same as that for any consistent and asymptotically invariant strategies when we consider the worst-case lower bound characterized by $\overline{\Delta}$.

## 4.2 The GNA-EBA Strategy

Based on the lower bounds, we design a strategy and show that its probability of misidentification aligns with the lower bounds. In the experimental design, we assume that the variances of outcomes are *known.*

Then, we propose the Generalized-Neyman-Allocation (GNA)-empirical-best-arm (EBA) strategy, which can be interpreted as a generalization of the Neyman allocation proposed by Neyman (1934). The pseudo-code is shown in Algorithm 1.

**Allocation rule: Generalized Neyman Allocation (GNA).** First, we define a target allocation ratio, which is used to determine our allocation rule, as follows:

$$w^{\text{GNA}} = \arg\max_{w \in \mathcal{W}} \min_{a^* \in [K], a \in [K] \setminus \{a^*\}} \frac{1}{2\Omega^{a^*,a}(w)},$$

which is identical to that in equation 6. Then, we allocate arms to experimental units as follows:

$$A_t = \begin{cases} 1 & \text{if} \quad t \leq \lceil w^{\text{GNA}}(1)T \rceil \\ 2 & \text{if} \quad \lceil w^{\text{GNA}}(1)T \rceil < t \leq \lceil \sum_{b=1}^{2} w^{\text{GNA}}(b)T \rceil \\ \vdots \\ K & \text{if} \quad \lceil \sum_{b=1}^{K-1} w^{\text{GNA}}(b)T \rceil < t \leq T \end{cases}.$$

**Recommendation rule: Empirical Best Arm (EBA).** After the final round $T$, we recommend $\widehat{a}_T \in [K]$, an estimate of the best arm, defined as

$$\widehat{a}_T^{\text{EBA}} = \arg\max_{a \in [K]} \widehat{\mu}_T^a, \qquad \widehat{\mu}_T^a = \frac{1}{\lceil w^{\text{GNA}}(a)T \rceil} \sum_{t=1}^{T} \mathbb{1}[A_t = a]Y_t. \tag{3}$$

Our strategy generalizes the Neyman allocation because for $w^{\text{GNA}}$ in the GNA allocation rule, $w^{\text{GNA}} = \left( \frac{\sigma^1}{\sigma^1 + \sigma^2}, \frac{\sigma^2}{\sigma^1 + \sigma^2} \right)$ when $K = 2$, which is a target allocation ratio of the Neyman allocation. The EBA recommendation rule is one of the typical recommendation rules and used in other strategies in fixed-budget BAI, such as the Uniform-EBA strategy (Bubeck et al., 2009; 2011).

### 4.3 Probability of Misidentification of the GNA-EBA strategy

This section shows an upper bound for the probability of misidentification of the GNA-EBA strategy. The proof is shown in Appendix E.

**Theorem 4.2** (Upper Bound of the GNA-EBA strategy)**.** *For any $0 < \underline{\Delta} \leq \overline{\Delta} < \infty$, any $a^* \in [K]$, and any $P_0 \in \mathcal{P}^{\text{G}}(a^*, \underline{\Delta}, \overline{\Delta})$, the GNA-EBA strategy satisfies*

$$\liminf_{T \to \infty} -\frac{1}{T} \log \mathbb{P}_{P_0} \left( \widehat{a}_T^{\text{EBA}} \neq a^* \right) \geq \frac{\underline{\Delta}^2}{2\Omega^{a^*,a}(w^{\text{GNA}})}.$$

Then, the worst-case upper bound is given as the following theorem.

**Theorem 4.3** (Worst-case upper bound of the GNA-EBA strategy)**.** *For any $0 < \underline{\Delta} \leq \overline{\Delta} < \infty$, the GNA-EBA strategy satisfies*

$$\inf_{P \in \cup_{a^* \in [K]} \mathcal{P}^{\text{G}}(a^*, \underline{\Delta}, \overline{\Delta})} \liminf_{T \to \infty} -\frac{1}{T} \log \mathbb{P}_P \left( \widehat{a}_T^{\text{EBA}} \neq a^* \right) \geq \text{UpperBound}(\underline{\Delta}) := \min_{a^* \in [K]} \frac{\underline{\Delta}^2}{2\Omega^{a^*,a}(w^{\text{GNA}})} = \max_{w \in \mathcal{W}} \min_{a^* \in [K]} \frac{\underline{\Delta}^2}{2\Omega^{a^*,a}(w)}.$$

*Proof.* By taking $\min_{a^* \in [K]} \inf_{P \in \mathcal{P}^{\text{G}}(a^*, \underline{\Delta}, \overline{\Delta})}$ in both LHS and RHS in the upper bound of Theorem 4.2, we obtain the statement. $\square$

### 4.4 Worst-case Asymptotic Optimality

By comparing the lower bound in Theorem 4.1 and the upper bound in Theorem 4.3, the following theorem shows that they approach as $\underline{\Delta} - \overline{\Delta} \to 0$ and match when $\underline{\Delta} = \overline{\Delta}$. This case implies a situation in which the difference between the expected outcome of the best arm and the next best arm is equal to the difference between the expected outcome of the best arm and the worst arm. In such a situation, it is possible to construct an optimal algorithm even if the expected outcomes are unknown. This concept is illustrated in Figure 1.

**Theorem 4.4.** *As $\underline{\Delta} - \overline{\Delta} \to 0$, we have*

$$\mathrm{LowerBound}(\overline{\Delta}) - \mathrm{UpperBound}(\underline{\Delta}) \to 0.$$

*Additionally, for any $0 < \Delta < \infty$ and for $\overline{\mathcal{P}}^{\mathrm{G}}(\Delta) \coloneqq \left\{ P \in \mathcal{P}^{\mathrm{G}}(a^*, \underline{\Delta}, \overline{\Delta}) \mid a^* \in [K], \ \ \underline{\Delta} = \overline{\Delta} = \Delta \right\}$, we have*

$$\sup_{\pi \in \Pi^{\mathrm{cons}}} \inf_{P \in \overline{\mathcal{P}}^{\mathrm{G}}(\Delta)} \liminf_{T \to \infty} -\frac{1}{T} \log \mathbb{P}_P \left( \widehat{a}_T^{\mathrm{EBA}} \neq a^*(P) \right) \leq \max_{w \in \mathcal{W}} \min_{a^* \in [K]} \frac{\Delta^2}{2\Omega^{a^*, a}(w)}$$

$$= \max_{w \in \mathcal{W}} \inf_{P \in \overline{\mathcal{P}}^{\mathrm{G}}(\Delta)} \Gamma(P, w) \leq \inf_{P \in \overline{\mathcal{P}}^{\mathrm{G}}(\Delta)} \liminf_{T \to \infty} -\frac{1}{T} \log \mathbb{P}_P \left( \widehat{a}_T^{\mathrm{EBA}} \neq a^*(P) \right).$$

*Proof.* From Theorem 4.1, we have $\inf_{P \in \cup_{a^* \in [K]} \mathcal{P}^{\mathrm{G}}(a^*, \underline{\Delta}, \overline{\Delta})} \limsup_{T \to \infty} -\frac{1}{T} \log \mathbb{P}_P(\widehat{a}_T^\pi \neq a^*) \leq \mathrm{LowerBound}(\overline{\Delta})$. From Theorem 4.3, we have $\inf_{P \in \cup_{a^* \in [K]} \mathcal{P}^{\mathrm{G}}(a^*, \underline{\Delta}, \overline{\Delta})} \liminf_{T \to \infty} -\frac{1}{T} \log \mathbb{P}_P \left( \widehat{a}_T^{\mathrm{EBA}} \neq a^* \right) \geq \mathrm{UpperBound}(\underline{\Delta})$.

From the definitions of $\mathrm{LowerBound}(\overline{\Delta})$ and $\mathrm{UpperBound}(\underline{\Delta})$, we directly have $\mathrm{LowerBound}(\overline{\Delta}) - \mathrm{UpperBound}(\underline{\Delta}) \to 0$ as $\overline{\Delta} - \underline{\Delta} \to 0$.

Furthermore, if $\overline{\Delta} = \underline{\Delta} = \Delta$ holds, then $\mathrm{LowerBound}(\overline{\Delta}) = \mathrm{UpperBound}(\underline{\Delta}) = \max_{w \in \mathcal{W}} \min_{a^* \in [K]} \frac{\Delta^2}{2\Omega^{a^*, a}(w)}$ holds. Note that $\max_{w \in \mathcal{W}} \min_{a^* \in [K]} \frac{\Delta^2}{2\Omega^{a^*, a}(w)} = \inf_{P \in \overline{\mathcal{P}}^{\mathrm{G}}(\Delta)} \max_{w \in \mathcal{W}}, \Gamma(P, w)$ holds (see Section 5.5). Thus, the proof completes. □

This result implies that $\inf_{P \in \overline{\mathcal{P}}^{\mathrm{G}}(\Delta)} \max_{w \in \mathcal{W}} \Gamma^*(P, w)$ **is lower and upper bounds for any consistent strategies (** $\mathrm{LowerBound}(\overline{\Delta}) = \mathrm{UpperBound}(\underline{\Delta}) = \inf_{P \in \overline{\mathcal{P}}^{\mathrm{G}}(a^*, \Delta)} \max_{w \in \mathcal{W}} \Gamma^*(P, w)$**), although there exists $P \in \mathcal{P}^{\mathrm{G}}_{\sigma=1}$ such that a lower bound is larger than** $\max_{w \in \mathcal{W}} \Gamma^*(P, w)$ **(Degenne, 2023).** Thus, without contradicting the existing results by Ariu et al. (2021) and Degenne (2023), we show the existence of an optimal strategy for a lower bound conjectured by Kaufmann et al. (2016) and Garivier & Kaufmann (2016). We illustrate this result in Figure 2. Our findings do not present a contradiction to the results by Ariu et al. (2021) and Degenne (2023) because we restrict the bandit models.

Figure 1: An idea in the derivation of the lower bounds. To lower bound the probability of misidentification (upper bound $-\frac{1}{T} \log \mathbb{P}_P(\widehat{a}_T^\pi \neq a^*(P))$) it is sufficient to consider a case in the right figure.

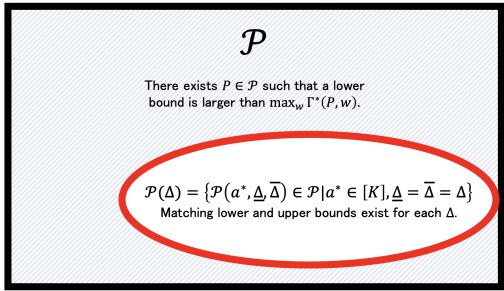

Figure 2: The region where there exists matching lower and upper bounds.

Additionally, when $\underline{\Delta} = \overline{\Delta}$ and the variances are known, we design an asymptotically worst-case optimal strategy, which is *not* based on the uniform allocation.

Note that our optimality criterion is closely related to local optimality that has been used in various studies, including the bandit problem and statistical testing. For example, the optimality under a small-gap setting such that $\mu^{a^*(P)}(P) - \mu^a(P) \to 0$ for all $a \in [K] \backslash \{a^*\}$ is a particular case of our optimality. Such a small-gap optimality has been discussed in Jamieson et al. (2014) and Shin et al. (2018), where the former is a study about fixed-confidence BAI, and the latter is about fixed-budget BAI. The local Bahadur efficiency is also a closely related optimality criterion, which has been used in the literature of statistical testing (Wieand, 1976; Akritas & Kourouklis, 1988; He & Shao, 1996). In many cases, such a location has been employed to discuss the optimality of the main performance measure while ignoring estimation errors of other parameters.

From the viewpoints of open questions, we answer them as follows:

- For Open questions 1 and 3: we prove that there exist matching lower and upper bounds under the worst case metric $(\inf_{P \in \cup_{a^* \in [K]} \mathcal{P}^G(a^*, \underline{\Delta}, \overline{\Delta})} \limsup_{T \to \infty} -\frac{1}{T} \log \mathbb{P}_P(\widehat{a}_T^\pi \neq a^*(P)))$ and the restricted bandit models $(\overline{\Delta} = \underline{\Delta})$. Additionally, even if we restrict the class to asymptotically invariant strategies, the lower bound cannot be improved.

- For Open question 2: we prove that a variance-based strategy performs better than the uniform allocation in Gaussian bandit models.

As a side-product of our study, we propose a novel setting called hypothesis BAI (HBAI) in Section 6. The results shown above consider a lower bound that is valid for any choices of the mean parameters and the best arm, reflecting a situation where they are unknown. However, we can consider a situation where there is a conjecture of the best arm prior to an experiment. For example, as well as hypothesis testings, we can set null and alternative hypotheses such that a conjectured best arm $\widetilde{a} \in [K]$ is truly best (alternative) or not (null). Under this setting, if we set $\widetilde{a}$ as a proxy of $a^*$ and conduct an experiment, then the probability of minimization is minimized when $\widetilde{a} = a^*$ (the alternative hypothesis is true). As an analogy of hypothesis testing, we call this setting HBAI.

Results of simulation studies are shown in Section 7. We discuss related work in Appendix A. In Appendix 5, we explain intuitive ideas about the lower bounds, which include the preliminary of the proof of Theorem 4.1.

### 4.5 Discussion

Finally, we discuss our results in comparison to existing results. This section aims to clarify our study's contribution by showing how we approach the open question raised in existing studies.

**Comparison with Komiyama et al. (2022) and Degenne (2023).** First, we compare our results with those in Komiyama et al. (2022) and Degenne (2023). While Komiyama et al. (2022) and Degenne (2023) measure a performance by $\inf_{P \in \mathcal{P}} H(P) \limsup_{T \to \infty} -\frac{1}{T} \log \mathbb{P}_P(\widehat{a}_T^\pi \neq a^*)$, we measure it by $\inf_{P \in \mathcal{P}} \limsup_{T \to \infty} -\frac{1}{T} \log \mathbb{P}_P(\widehat{a}_T^\pi \neq a^*)$, where the main difference is the existence of the problem difficulty term $H(P)$. Degenne (2023) shows a lower bound $\inf_{P \in \mathcal{P}} H(P) \liminf_{T \to \infty} -\frac{1}{T} \log \mathbb{P}_P(\widehat{a}_T \neq a^*) \leq \max_{w \in \mathcal{W}} \Lambda^*(P, w, H)$, which is tighter than the lower bound in Komiyama et al. (2022). We compare this lower bound to ours. Let us consider a case $H(P)$ is constant independent of $P$ and $T$; that is, $H(P) = \overline{H}$ holds for a constant $\overline{H}$ independent of $P$ and $T$. Then, from Degenne (2023)'s lower bound, we have $\inf_{P \in \mathcal{P}} \liminf_{T \to \infty} -\frac{1}{T} \log \mathbb{P}_P(\widehat{a}_T \neq a^*) \leq \inf_{\mu^a < v^a < \mu^{a^*(P)}} \sum_{a \in [K]} w(a) \widetilde{\mathrm{KL}}(v^a, \mu^a(P)) \leq \mathrm{LowerBound}(\overline{\Delta})$, which aligns with our lower bound. This difference comes from how we interpret the difficulty of the problem of interest. Degenne (2023) considers a lower bound under $\mathcal{P}$, instead of introducing some difficulty term $H(P)$. In contrast, we consider a lower bound that matches the upper bound when $\overline{\Delta} - \underline{\Delta} \to 0$, which implies that we restrict the class of $\mathcal{P}$ when considering the asymptotic optimality.

**Asymptotically invariant Strategy.** In fixed-budget BAI, the strategy proposed by Glynn & Juneja (2004) has been considered a candidate for asymptotically optimal strategies. We and Degenne (2023) show

that it is actually an asymptotically optimal strategy among a class of consistent and asymptotically invariant (static) strategies. Degenne (2023) proves that there exists $P \in \mathcal{P}$ under which a lower bound for consistent strategies does not match that for consistent and asymptotically invariant (static) strategies. In contrast, in the following corollary, we show that under the worst-case analysis, the lower bound for consistent strategy is the same as that for consistent and asymptotically invariant (static) strategies, and the upper bound of the GNA-EBA strategy matches the lower bound. This result implies that (i) we cannot obtain a strategy with a smaller probability of misidentification even if relaxing the class of strategies, and (ii) there exists an asymptotically optimal strategy, at least if we know variances. This result challenges the conclusion of Degenne (2023), which states that there exists *no* optimal strategy in the sense that its upper bound aligns with that of asymptotically invariant strategies. Our result does not contradict to his result because we restrict the bandit models $\mathcal{P}$, rather than the strategy class.

**Corollary 4.5.** *For any $0 < \Delta < \infty$ any $\mathcal{P}^{\mathrm{G}}(a^*, \Delta, \Delta$, any consistent strategies have a lower bound* $\mathrm{LowerBound}(\overline{\Delta})$ *same as any consistent and asymptotically invariant strategies, which aligns with an upper bound* $\mathrm{UpperBound}(\underline{\Delta})$ *of the GNA-EBA strategy.*

**Matching lower and upper bounds.** Instead of considering restricted bandit models $(\overline{\Delta} - \underline{\Delta} \to 0)$, we develop tight lower and upper bounds that match as $\overline{\Delta} - \underline{\Delta} \to 0$. Degenne (2023) considers a lower bound characterized by a problem difficulty terms, where there is a gap from upper bounds. In other words, we derive matching lower and upper bounds for restricted bandit models under $\overline{\Delta} - \underline{\Delta} \to 0$, while Degenne (2023) derives a lower bound whose leading factor aligns with an upper bound (but it is not a perfect match) for global bandit models $\mathcal{P}$ (there is no restriction such that $\overline{\Delta} - \underline{\Delta} \to 0$).

**Applications of interest.** Thus, there are two frameworks, asymptotic optimality for restricted bandit models and that depending on the problem difficulty. The former is closely related to large-deviation-based asymptotically optimal strategies such as Glynn & Juneja (2004), while the latter is closely related to the standard fixed-budget BAI strategies such as the successive halving. The remaining question is which framework is appropriate, and it depends on applications. The asymptotic optimality for restricted bandit models is often considered in epidemiology and economics, where the number of arms is not large, and we need to identify the best arm among the other arms whose expected outcomes are close to but less than that of the best arm. For example, van der Laan (2008) and Hahn et al. (2011) consider local models, where parameters have $1/\sqrt{T}$-perturbation (van der Vaart, 1998), and discuss asymptotic optimality in such models. In contrast, the use of optimality based on the problem difficulty is appropriate for recommendation systems and online advertisements, where the number of arms is significantly large. In this case, as well as the successive halving, it is desirable to remove suboptimal arms as early as possible, depending on how fast and accurately we can remove them depending on the problem difficulty.

## 5   Intuition behind Lower Bounds and the Proofs of Theorem 4.1 and Corollary 4.5

This section provides intuitive ideas behind Theorem 4.1 and Corollary 4.5 with preliminaries for their proofs. Our arguments about lower bounds are based on the information-theoretical approach. We reveal that different available information yields different lower bounds. Finally, we develop the worst-case lower bound for multi-armed Gaussian bandits.

### 5.1   Existence of Asymptotically Optimal Strategies

The existence of asymptotically optimal strategies is a longstanding open question (Kaufmann, 2020; Ariu et al., 2021; Degenne, 2023). Kaufmann et al. (2016) provide a lower bound and an asymptotically optimal strategy for two-armed Gaussian bandits with known variances. However, when the number of arms is three or more ($K \geq 3$), even the lower bounds remain unknown (see Table 1).

There are multiple reasons why this problem is challenging, and it is difficult to offer a single clear explanation. For instance, factors affecting the upper bounds of strategies include (i) the estimation error of distributional information, (ii) the class of strategies, and (iii) the dependency of optimal sample allocation ratios on the best arm.

To address this problem, we develop novel lower bounds by extending those shown by Kaufmann et al. (2016), focusing on the lack of distributional information.

## 5.2 Transportation Lemma

Our arguments about lower bounds start from a (conjectured) lower bound shown by Kaufmann et al. (2016) and Garivier & Kaufmann (2016).

**Lemma 5.1** (Lower bound given known distributions). *For any $0 < \underline{\Delta} \leq \overline{\Delta} < \infty$, any $a^* \in [K]$, and any $P_0 \in \mathcal{P}^{\mathrm{G}}(a^*, \underline{\Delta}, \overline{\Delta})$, any consistent (Definition 3.1) strategy $\pi \in \Pi^{\mathrm{cons}}$ satisfies*

$$\limsup_{T \to \infty} -\frac{1}{T} \log \mathbb{P}_{P_0}(\widehat{a}_T^\pi \neq a^*) \leq \limsup_{T \to \infty} \inf_{Q \in \cup_{b \neq a^*} \mathcal{P}(b, \underline{\Delta}, \overline{\Delta})} \sum_{a \in [K]} \kappa_{T,Q}^\pi(a) \mathrm{KL}(Q^a, P_0^a).$$

Here, $Q$ is an alternative hypothesis that is used for deriving lower bounds and not an actual distribution. Note that upper (resp. lower) bounds for $-\frac{1}{T} \log \mathbb{P}_{P_0}(\widehat{a}_T^\pi \neq a^*)$ corresponds to lower (resp. upper) bounds for $\mathbb{P}_{P_0}(\widehat{a}_T^\pi \neq a^*)$.

For two-armed Gaussian bandits, the lower bound can be simplified (See Theorem 12 in Kaufmann et al. (2016)). In this case, it is known that by allocating arm 1 and 2 with sample sizes $\frac{\sigma^1}{\sigma^1 + \sigma^2} T$ and $\frac{\sigma^2}{\sigma^1 + \sigma^2} T$, we can design asymptotically optimal strategy. Strategies using this allocation rule are called the *Neyman allocation* (Neyman, 1934).

## 5.3 Lower Bound given Known Distributions

However, when $K \geq 3$, there occur several issues in the derivation of lower bounds, and there does not exist an asymptotically optimal strategy whose upper bound aligns with the conjectured lower bound in Lemma 5.1 (Kaufmann, 2020; Ariu et al., 2021; Degenne, 2023). One of the difficulties comes from the fact that the term $\kappa_{T,Q}^\pi(a)$ does not correspond to sample allocation under the DGP $P_0$ (Kaufmann, 2020), which incurs a problem called the reverse KL problem (Kaufmann, 2020).

To derive lower bounds, we consider restricting strategies to ones such that the limit of $\kappa_{T,P}^\pi(a)$ ($\lim_{T \to \infty} \kappa_{T,P}^\pi(a)$) is the same across $P \in \mathcal{P}^{\mathrm{G}}(a^*, \underline{\Delta}, \overline{\Delta})$. Such a class of strategies are defined as asymptotically invariant strategies in Definition 3.2. Note that $\kappa_{T,P}^\pi$ is a deterministic value without randomness because it is an expected value of $\frac{1}{T} \sum_{t=1}^T \mathbb{1}[A_t^\pi = a]$. A typical example of this class of strategies is one using uniform allocation, such as the Uniform-EBA strategy (Bubeck et al., 2011). Another example is a strategy using the allocation rule only based on variances, such as the Neyman allocation (Neyman, 1934).

Given an asymptotically invariant strategy $\pi$, there exists $w^\pi \in \mathcal{W}$ such that for all $P \in \mathcal{P}^{\mathrm{G}}(a^*, \underline{\Delta}, \overline{\Delta})$, and $a \in [K]$, $\left| w^\pi(a) - \frac{1}{T} \sum_{t=1}^T \mathbb{E}_P [\mathbb{1}[A_t = a]] \right| \to 0$ holds.

Therefore, for any consistent and asymptotically invariant strategy $\pi$, the following lower bounds hold. The proof is shown in Appendix B.

**Lemma 5.2** (Lower bound given known distributions). *For any $0 < \underline{\Delta} \leq \overline{\Delta} < \infty$, any $a^* \in [K]$, and any $P_0 \in \mathcal{P}^{\mathrm{G}}(a^*, \underline{\Delta}, \overline{\Delta})$, any consistent (Definition 3.1) and asymptotically invariant (Definition 3.2) strategy $\pi \in \Pi^{\mathrm{cons}} \cap \Pi^{\mathrm{inv}}$ satisfies*

$$\limsup_{T \to \infty} -\frac{1}{T} \log \mathbb{P}_{P_0}(\widehat{a}_T^\pi \neq a^*) \leq \sup_{w \in \mathcal{W}} \min_{a \in [K] \setminus \{a^*\}} \frac{(\Delta^a(P_0))^2}{2\Omega^{a^*, a}(w)}.$$

We refer to a limit of the average sample allocation deduced from lower bounds as the *target allocation ratio* and denote it by $w^*$. We can derive various $w^*$ in different lower bounds. For example, in Lemma 5.2, because $\max_{w \in \mathcal{W}} \min_{a \in [K] \setminus \{a^*\}} \inf_{\substack{(\mu^b) \in \mathbb{R}^K \\ \mu^a > \mu^{a^*}}} w(a) \frac{(\mu^a - \mu^a(P_0))^2}{2(\sigma^a)^2}$ exists, we define the target allocation ratio as

$$w^* = \arg\max_{w \in \mathcal{W}} \min_{a \in [K] \setminus \{a^*\}} \frac{(\Delta^a(P_0))^2}{2\Omega^{a^*, a}(w)}. \tag{4}$$

The target allocation ratio $w^*$ works as a candidate about optimal sample allocation of optimal strategies. Here, note that the average sample allocation ratio is linked to an actual strategy, and we can compute $w^*$ independently of each instance $P_0$.

For the asymptotically invariant strategy, we can show that the strategy proposed by Glynn & Juneja (2004) is feasible if we can compute $\mathrm{KL}(Q^a, P_0^a)$, and under the strategy, the probability of misidentification aligns with the lower bound with asymptotically invariant strategies. This result is also shown by Degenne (2023) for more general distributions.

## 5.4 Uniform Lower Bound

As discussed by Glynn & Juneja (2004) and us, when we know distributional information completely, we can obtain an asymptotically optimal strategy whose probability of misidentification matches the lower bounds in Lemma 5.2. However, when we do not have full information, the strategy by Glynn & Juneja (2004) is infeasible. Furthermore, there exists $P_0 \in \mathcal{P}^{\mathrm{G}}(a^*, \underline{\Delta}, \overline{\Delta})$ whose lower bound is larger than that of Kaufmann et al. (2016).

For example, in Lemma 5.2, the target allocation ratio is given as equation 4. However, the target allocation ratio depends on unknown mean parameters $\mu^a(P_0)$ and the true best arm $a^*$. Therefore, strategies using the target allocation ratio are infeasible if we do not know those values.

We elucidate this problem by examining how our available information affects the lower bounds. Specifically, we consider lower bounds that are uniformly valid regardless of the missing information. First, we consider characterizing the lower bound in Lemma 5.2 by $\overline{\Delta}$, an upper bound of $\Delta^a(P_0)$ as the following lemma.

**Theorem 5.3** (Uniform lower bound)**.** *For any $0 < \underline{\Delta} \le \overline{\Delta} < \infty$, any $a^* \in [K]$, and any $P_0 \in \mathcal{P}^{\mathrm{G}}(a^*, \underline{\Delta}, \overline{\Delta})$, any consistent (Definition 3.1) and asymptotically invariant (Definition 3.2) strategy $\pi \in \Pi^{\mathrm{cons}} \cap \Pi^{\mathrm{inv}}$ satisfies*

$$\limsup_{T \to \infty} -\frac{1}{T} \log \mathbb{P}_{P_0}(\widehat{a}_T^\pi \ne a^*) \le \frac{\overline{\Delta}^2}{2\left(\sigma^{a^*} + \sqrt{\sum_{a \in [K] \setminus \{a^*\}} (\sigma^a)^2}\right)^2}.$$

The proof is shown in Appendix C. Here, the target allocation ratio is given as

$$
\begin{aligned}
w^*(a^*) &= \frac{\sigma^{a^*}}{\sigma^{a^*} + \sqrt{\sum_{b \in [K] \setminus \{a^*\}} (\sigma^b)^2}}, \\
w^*(a) &= \frac{(\sigma^b)^2 / \sqrt{\sum_{b \in [K] \setminus \{a^*\}} (\sigma^b)^2}}{\sigma^{a^*} + \sqrt{\sum_{b \in [K] \setminus \{a^*\}} (\sigma^b)^2}} = (1 - w^*(a^*)) \frac{(\sigma^a)^2}{\sum_{b \in [K] \setminus \{a^*\}} (\sigma^b)^2}, \quad \forall a \in [K] \setminus \{a^*\}.
\end{aligned}
\tag{5}
$$

Note that the lower bounds are characterized by the variances and the true best arm $a^*$. If we know them, we can design optimal strategies whose upper bound aligns with these lower bounds.

When designing strategies, variances and the best arm are required to construct the target allocation ratio. However, assuming that the best arm $a^*$ is known is unrealistic. We also cannot estimate it during an experiment because such a strategy violates the assumption of asymptotically invariant strategies. For example, Shin et al. (2018) estimates $a^*$ during an experiment under the framework of Glynn & Juneja (2004), but such a strategy does not satisfy the asymptotically invariant strategies because $a^*$ and $w^*$ can differ across the choice of $P_0$. When considering asymptotically invariant strategies, we need to know $a^*$ before starting an experiment.

To circumvent this issue, one approach is to fix $\widetilde{a}$, independent of $P_0$, before an experiment. We then construct a target allocation ratio as $w^\dagger(\widetilde{a}) = \frac{\sigma^{\widetilde{a}}}{\sigma^{\widetilde{a}} + \sqrt{\sum_{b \in [K] \setminus \widetilde{a}} (\sigma^b)^2}}$ and $w^\dagger(a) = (1 - w^\dagger(\widetilde{a})) \frac{(\sigma^a)^2}{\sum_{b \in [K] \setminus \widetilde{a}} (\sigma^b)^2}$ for all $a \in [K] \setminus \widetilde{a}$. arms are then allocated following this target allocation ratio. Under a strategy using such an allocation rule, if $\widetilde{a}$ equals $a^*$, the target allocation ratio $w^\dagger$ aligns with $w^*$ in equation 5. We refer to

this setting as *hypothesis BAI* (HBAI), as the formulation resembles hypothesis testing. For details, see Section 6.

However, when $\widetilde{a}$ is *not* equal to $a^*$, the target allocation ratio $w^\dagger$ does not equal $w^*$, under which the strategy becomes suboptimal. In the following section, we consider a best-arm agnostic lower bound by contemplating the worst-case scenario for the choice of $a^*$.

**Remark 1.** When $K = 2$, the target allocation ratio has a closed-form such that $w^{\mathrm{GNA}}(a) = \frac{\sigma^a}{\sigma^1 + \sigma^2}$ for $a \in [K] = \{1, 2\}$. Note that the target allocation ratio becomes independent of $a^*$ in this case. Allocation rules following this target allocation ratio are referred to as the Neyman allocation (Neyman, 1934).

### 5.5 Best-Arm-Worst-Case Lower Bound

As discussed above, $a^*$ is unknown in practice, and we consider the worst-case analysis regarding the choice of the best arm. We represent the worst-case for all possible best arms by using the following metric:

$$\inf_{P_0 \in \cup_{a^* \in [K]} \mathcal{P}^{\mathrm{G}}(a^*, \underline{\Delta}, \overline{\Delta})} \limsup_{T \to \infty} -\frac{1}{T} \log \mathbb{P}_{P_0}(\widehat{a}_T^\pi \neq a^*(P)).$$

Thus, by taking the worst case over $P_0$, we obtain the following lower bound.

**Theorem 5.4** (Best-arm-worst-case lower bound)**.** *For any $0 < \underline{\Delta} \leq \overline{\Delta} < \infty$, any consistent (Definition 3.1) and asymptotically invariant (Definition 3.2) strategy $\pi \in \Pi^{\mathrm{cons}} \cap \Pi^{\mathrm{inv}}$ satisfies*

$$\inf_{P_0 \in \cup_{a^* \in [K]} \mathcal{P}^{\mathrm{G}}(a^*, \underline{\Delta}, \overline{\Delta})} \limsup_{T \to \infty} -\frac{1}{T} \log \mathbb{P}_{P_0}(\widehat{a}_T^\pi \neq a^*(P_0)) \leq \mathrm{LowerBound}(\overline{\Delta}).$$

*Proof.* From Lemma 5.2, for each $w \in \mathcal{W}$, we have

$$\limsup_{T \to \infty} -\frac{1}{T} \log \mathbb{P}_{P_0}(\widehat{a}_T^\pi \neq a^*(P_0)) \leq \frac{\overline{\Delta}^2}{2\Omega^{a^*, a}(w)}.$$

Therefore, by taking $\min_{a^* \in [K]}$ in both LHS and RHS, we have

$$\min_{a^* \in [K]} \inf_{P_0 \in \mathcal{P}^{\mathrm{G}}(a^*, \underline{\Delta}, \overline{\Delta})} \limsup_{T \to \infty} -\frac{1}{T} \log \mathbb{P}_{P_0}(\widehat{a}_T^\pi \neq a^*(P_0)) \leq \min_{a^* \in [K]} \inf_{P \in \mathcal{P}^{\mathrm{G}}(a^*, \underline{\Delta}, \overline{\Delta})} \frac{\overline{\Delta}^2}{2\Omega^{a^*, a}(w)}.$$

By taking the maximum over $w$, we complete the proof. $\square$

The target allocation ratio deduced from this lower bound is

$$w^* = \arg\max_{w \in \mathcal{W}} \min_{a^* \in [K], \, a \in [K] \setminus \{a^*\}} \frac{1}{2\Omega^{a^*, a}(w)}. \tag{6}$$

Here, note that $\max_{w \in \mathcal{W}} \max_{a^* \in [K]} \min_{a \in [K] \setminus \{a^*\}} \frac{\overline{\Delta}^2}{2\Omega^{a^*, a}(w)}$ does not have a closed-form solution and requires numerical computations.

Note that when $K = 2$, the target allocation ratio has a closed-form solution such that $w^*(a) = \frac{\sigma^a}{\sigma^1 + \sigma^2}$ for $a \in [2]$. Additionally, the lower bound is given as

$$\inf_{P_0 \in \cup_{a^* \in [K]} \mathcal{P}^{\mathrm{G}}(a^*, \underline{\Delta}, \overline{\Delta})} \limsup_{T \to \infty} -\frac{1}{T} \log \mathbb{P}_{P_0}(\widehat{a}_T^\pi \neq a^*(P)) \leq \frac{\overline{\Delta}^2}{2(\sigma^1 + \sigma^2)^2}.$$

This target allocation ratio and lower bound are equal to those in the lower bound for two-armed Gaussian bandits shown by Kaufmann et al. (2016).

The target allocation ratio is independent of $a^*$. Therefore, we can avoid the issue of dependency on $a^*$, which cannot be estimated in an experiment.

**Remark 2** (Inequalities of lower bounds). Note that for the derived lower bounds, $\text{LowerBound}(\overline{\Delta}) \leq \dfrac{\overline{\Delta}^2}{2\left(\sigma^{a^*} + \sqrt{\sum_{a \in [K] \setminus \{a^*\}} (\sigma^a)^2}\right)^2}$ holds. The larger lower bounds imply tighter lower bounds; that is, $\mathbb{P}_{P_0}(\widehat{a}_T^\pi \neq a^*)$ is smaller as the lower bounds become larger.

**Remark 3** (Two-armed Gaussian bandits). When $K = 2$, the above serial arguments about the lower bound can be simplified. The reason why we cannot use Theorem 5.3 is because the target allocation ratio depends on $a^*$. However, when $K = 2$, the target allocation ratio is independent of $a^*$ and given as the ratio of the standard deviations. This is because the comparison between the best and suboptimal arms plays an important role, which requires the best arm $a^*$. However, when $K = 2$, a pair of comparisons is unique; that is, we always allocate arms comparing arms 1 and 2, regardless of which arm is best. Therefore, we can simplify the lower bounds when $K = 2$. Specifically, optimal strategies just allocate treatment with the ratio of the standard deviation, which is also referred to as the Neyman allocation (Neyman, 1934). Also, see Theorem 12 in Kaufmann et al. (2016) for details.

## 5.6 Relaxing the Strategy Class Restrictions

In Theorem 5.4, we restrict the strategies to consistent and asymptotically invariant ones. However, the latter restriction can be removed. We introduced the restriction of asymptotically invariant strategies to derive a lower bound from $\limsup_{T\to\infty} -\frac{1}{T}\log \mathbb{P}_{P_0}(\widehat{a}_T^\pi \neq a^*) \leq \limsup_{T\to\infty} \inf_{Q \in \cup_{b \neq a^*} \mathcal{P}(b,\underline{\Delta},\overline{\Delta})} \sum_{a \in [K]} \kappa_{T,Q}^\pi(a) \text{KL}(Q^a, P_0^a)$ in Lemma 5.1. Although we discussed $\inf_{P_0 \in \cup_{a^* \in [K]} \mathcal{P}^{\text{G}}(a^*,\underline{\Delta},\overline{\Delta})} \limsup_{T\to\infty} -\frac{1}{T}\log \mathbb{P}_{P_0}(\widehat{a}_T^\pi \neq a^*(P))$ as a result of information-theoretical analysis, we can derive a lower bound from

$$\inf_{Q \in \cup_{a^* \in [K]} \mathcal{P}^{\text{G}}(a^*,\underline{\Delta},\overline{\Delta})} \limsup_{T\to\infty} -\frac{1}{T}\log \mathbb{P}_Q(\widehat{a}_T^\pi \neq a^*(Q)) \leq \limsup_{T\to\infty} \inf_{Q \in \cup_{b \neq a^*} \mathcal{P}(b,\underline{\Delta},\overline{\Delta})} \sum_{a \in [K]} \kappa_{T,P_0}^\pi(a) \text{KL}(P_0^a, Q^a).$$

This metric is mathematically equivalent to $\inf_{P_0 \in \cup_{a^* \in [K]} \mathcal{P}^{\text{G}}(a^*,\underline{\Delta},\overline{\Delta})} \limsup_{T\to\infty} -\frac{1}{T}\log \mathbb{P}_{P_0}(\widehat{a}_T^\pi \neq a^*(P))$, but we can derive a lower bound without using the asymptotically invariant strategy restriction. This is because $\kappa_{T,P_0}^\pi(a)$ appears in the lower bound, instead of $\kappa_{T,Q}^\pi(a)$.

We show the lower bound in Theorem 4.1 and the proof in Appendix D.

Additionally, from the lower bounds in Theorems 5.4 and 4.1 and the upper bound in Theorem 4.3, we obtain Corollary 4.5. This corollary implies that when $\overline{\Delta} = \underline{\Delta}$, we can obtain an asymptotically optimal strategy if we know variances.

## 6 Hypothesis BAI

Based on the arguments in Section 5.4, we design the Hypothesis GNA-EBA (H-GNA-EBA) strategy that utilizes a conjecture $\widetilde{a} \in [K]$ of $a^*$, instead of considering the worst-case for $a^*$. We refer to $\widetilde{a}$ as a hypothetical best arm.

In the H-GNA-EBA strategy, we define a target allocation ratio as $w^{\text{H-GNA}}(\widetilde{a}) = \dfrac{\sigma^{\tilde{a}}}{\sigma^{\tilde{a}} + \sqrt{\sum_{b \in [K] \setminus \{\tilde{a}\}} (\sigma^b)^2}}$ and $w^{\text{H-GNA}}(a) = \left(1 - w^{\text{H-GNA}}(\widetilde{a})\right) \dfrac{(\sigma^a)^2}{\sum_{b \in [K] \setminus \{\tilde{a}\}} (\sigma^b)^2}$ for all $a \in [K] \setminus \{\widetilde{a}\}$. If $\widetilde{a}$ is equal to $a^*$, the upper bound of the H-GNA-EBA strategy for the probability of misidentification aligns with the lower bound in Theorem 5.3.

This strategy is more suitable under a setting different from BAI, where there are null and alternative hypotheses such that $H_0 : a^* \neq \widetilde{a} \in [K]$ and $H_1 : a^* = \widetilde{a}$; that is, the null hypothesis corresponds to a situation where the hypothetical best arm is *not* the best. In contrast, the alternative hypothesis posits that the hypothetical best arm is the best. Then, we consider minimizing the probability of misidentification when the alternative hypothesis is true. This probability corresponds to power in hypothesis testing. Our aim is to minimize the misidentification probability when the null hypothesis is false, corresponding to the

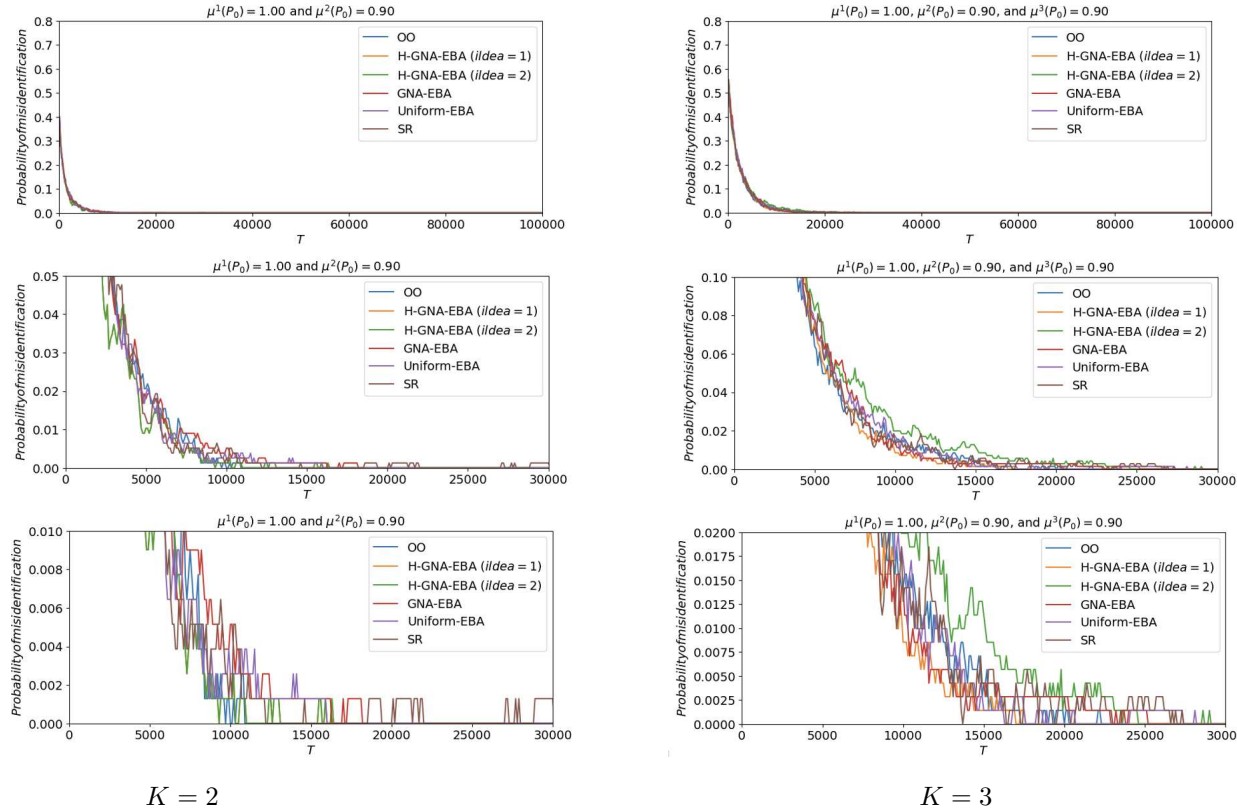

$K = 2$

$K = 3$

Figure 3: Experimental results ($K = 2$ and $K = 3$). The left figure shows the result with $K = 2$, and the right figure shows the result with $K = 2$. The $y$-axis and $x$-axis denote the probability of misidentification and $T$, respectively. We show the same results with different three scales of the $y$-axis and $x$-axis for each $K = 2$ and $K = 3$.

*power of the test.* We refer to this setting as Hypothesis BAI (HBAI). We present two examples for this setting.

**Example 1** (Online advertisement)**.** Let $\widetilde{a} \in [K]$ be an arm corresponding to a new advertisement. Our null hypothesis $a^* \neq \widetilde{a}$ implies that the existing advertisements $a \in [K]\backslash\{\widetilde{a}\}$ are superior to the new advertisement. Our goal is to reject the null hypothesis with a maximal probability when the null hypothesis is not correct; that is, the new hypothesis is better than the others.

**Example 2** (Clinical trial)**.** Let $\widetilde{a} \in [K]$ be a new drug. Our null hypothesis $a^* \neq \widetilde{a}$ implies that the existing drug $a \in [K]\backslash\{\widetilde{a}\}$ is superior to the new drug (equivalently, the new drug is not good as the existing drugs). Our goal is to reject the null hypothesis with a maximal probability when the new drug is better than the others.

The asymptotic efficiency of hypothesis testing is referred to as the *Bahadur efficiency* (Bahadur, 1960; 1967; 1971) of the test[10].

## 7 Simulation Studies

Using simulation studies, we investigate the performances of our GNA-EBA and the H-GNA-EBA strategies. We compare them with the Uniform-EBA strategy (Uniform, Bubeck et al., 2011), which allocates arms with the same allocation ratio ($1/K$), the successive rejects strategy (SR, Audibert et al., 2010), and the strategy

---

[10]Note that the Bahadur efficiency of a test evaluates $P$-values (random variable), not the probability of misidentification (non-random variable). However, these are closely related. See Bahadur (1967).

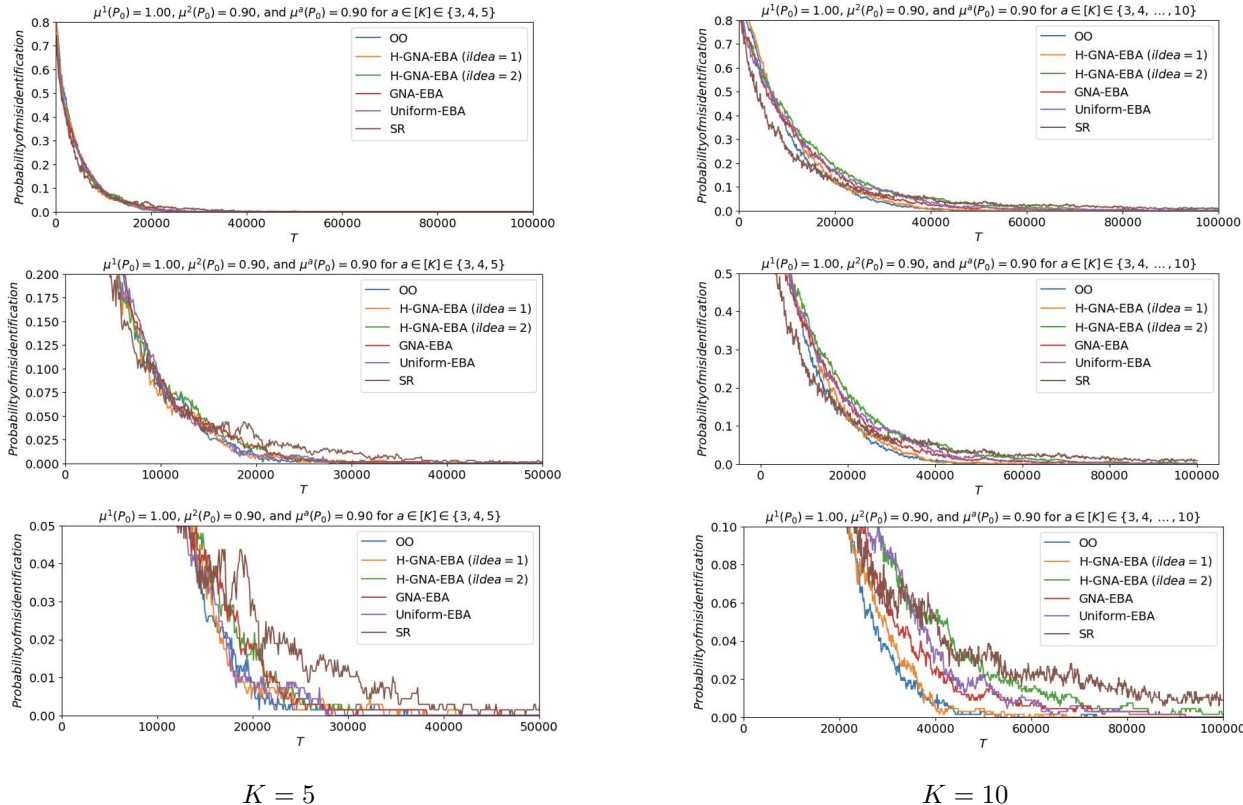

$K = 5$             $K = 10$

Figure 4: Experimental results ($K = 5$ and $K = 10$). The left figure shows the result with $K = 5$, and the right figure shows the result with $K = 10$. The $y$-axis and $x$-axis denote the probability of misidentification and $T$, respectively. We show the same results with different three scales of the $y$-axis and $x$-axis for each $K = 5$ and $K = 10$.

for ordinal optimization (OO) proposed by (Glynn & Juneja, 2004). Note that the OO requires full knowledge about distributions, including mean parameters and which arm is the best arm. However, if we know it, the OO yields the theoretically lowest probability of misidentification (Glynn & Juneja, 2004; Kaufmann, 2020; Degenne, 2023). Therefore, we regard the OO as a practically infeasible oracle strategy. We investigate two H-GNA-EBA strategies by setting $\widetilde{a} = 1$ and $\widetilde{a} = 2$. The SR strategy is often regarded as the most practical strategy.

Let $K \in \{2, 5, 10, 20\}$. The best arm is arm 1 and $\mu^1(P) = 1$. The expected outcomes of suboptimal arms 0.75 ($\mu^a(P) = 0.75$ for all $a \in [K]\backslash\{1\}$). The variances are drawn from a uniform distribution with support $[0.5, 5]$. We continue the strategies until $T = 100,000$. We conduct 100 independent trials for each setting. For each $T \in \{100, 200, 300, \cdots, 99,900, 100,000\}$, we plot the empirical probability of misidentification in Figures 3– 5. For each result, we show three figures with different scales on the $x$-axis and $y$-axis to observe the details.

Our theoretical results imply that the strategies with the highest performance are the OO and then the H-GNA-EBA strategy with $\widetilde{a} = 1$ (their performances are equal). Then, the GNA-EBA strategy and the Uniform strategy follow them. The other strategies follow these three strategies, but the order of the performances is not predictable from the theory.

From the results, as the theory predicts, strategies that use more information can achieve a lower probability of misidentification; that is, the OO and the H-GNA-EBA strategy with $\widetilde{a} = 1$ are the best strategies. Then, the GNA-EBA strategy follows them. Note that the OO and H-GNA-EBA strategies with $\widetilde{a} = 1$ are oracle strategies, which are feasible only when we can conjecture which arm is the best arm in advance.

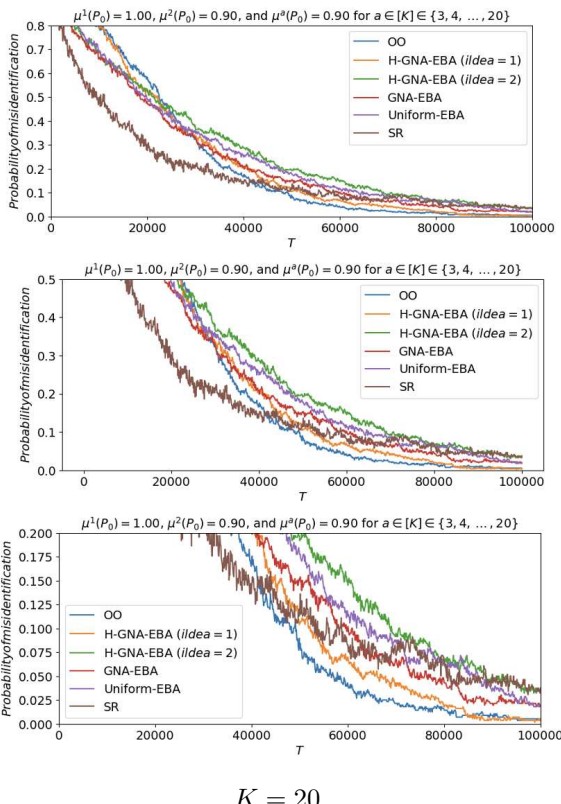

$K = 20$.

Figure 5: Experimental results ($K = 20$). The $y$-axis and $x$-axis denote the probability of misidentification and $T$, respectively. We show the same results with different three scales of the $y$-axis and $x$-axis.

Interestingly, the SR strategy performs well in early rounds, but the other asymptotically optimal strategies outperform it in late rounds.

In Appendix F, we conduct the additional three experiments. In the first experiment in the Appendix, the expected outcomes of suboptimal arms are drawn from a uniform distribution with support $[0.75, 0.90]$ for $a \in [K] \backslash \{1, 2\}$, while $\mu^2(P) = 0.75$. The variances are drawn from a uniform distribution with support $[0.5, 5]$. We continue the strategies until $T = 10,000$. We conduct 100 independent trials for each setting. For each $T \in \{100, 200, 300, \cdots, 99, 900, 100, 000\}$, we plot the empirical probability of misidentification in Figures 9– 18. In this case, our theoretical results imply that the highest performance is the OO, and then the H-GNA-EBA strategy with $\widetilde{a} = 1$, the GNA-EBA, and the Uniform strategies follow. The other strategies follow them, but the order of the performances is not predictable from the theory. In this experiment, the strategies show the predicted performances.

In the remaining two experiments, we conduct experiments with the same setting as the previous two experiments, but we changed the variances are drawn from a uniform distribution with support $[0.5, 10]$, while the variances are drawn from a uniform distribution with support $[0.5, 5]$ in the previous experiments. We plot the empirical probability of misidentification in Figures 3– 5. In those cases, our proposed strategies outperform the others due to the larger diversity of the variances. However, because the convergence also becomes slow, the SR outperforms in the early stages longer than the previous experiments, though our strategies outperform it in late rounds.

## 8 Conclusion

We investigated scenarios in experimental design, known as BAI or ordinal optimization. We found that the optimality of strategies significantly depends on the information available prior to an experiment. Based

on our findings, we developed the novel worst-case lower bounds for the probability of misidentification and then proposed a strategy whose worst-case probability of misidentification matches these worst-case lower bounds.

Lastly, we propose two important extensions of our results. First, we suggest the potential extension of our results to BAI with general distributions. In this study, we developed lower bounds for multi-armed Gaussian bandits. By approximating the KL divergence of distributions in a broader class under the small-gap regime, we can apply our results to various distributions, including Bernoulli distributions and general nonparametric models. In fact, Kato et al. (2023b) extends our results, developing lower bounds for more general distributions using semiparametric theory (Bickel et al., 1998; van der Vaart, 1998). These extended results, which refine those of Kato et al. (2023b), will be published later. It is important to note that in the small-gap regime, asymptotically optimal strategies for one-parameter bandit models, such as Bernoulli distributions, utilize uniform allocation because variances of outcomes become equal as gaps approach zero[11]. The results from Kato et al. (2023b) align with the conjecture in Kaufmann et al. (2016). From the results of Kato et al. (2023b), we can interpret the GNA-EBA strategy as an extension of the Uniform-EBA strategy proposed by Bubeck et al. (2009; 2011), which recommends uniform allocation in bandit models with bounded supports.

The second extension is optimal strategies that estimate variances during an experiment, instead of assuming known variances. If variances are estimated during an experiment, the estimation error affects the probability of misidentification. We conjecture that an optimal strategy exists that estimates variances during an experiment because similar results can be obtained under the central limit theorem (van der Laan, 2008; Hahn et al., 2011; Hadad et al., 2021; Kato et al., 2020; 2023a). However, its existence remains unproven due to the difficulty in evaluating tail probabilities. Related work, such as that by Kato et al. (2023a) and Jourdan et al. (2023), may help to resolve this issue.

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

# A    Related Work

This section introduces related work.

## A.1    Historical Background of Ordinal Optimization and BAI

Researchers have acknowledged the importance of statistical inference and experimental approaches as essential scientific tools (Peirce & Jastrow, 1884; Peirce & de Waal, 1887). With the advancement of these statistical methodologies, experimental design also began attracting attention. Fisher (1935) develops the groundwork for the principles of experimental design. Wald (1949) establishes fundamental theories for statistical decision-making, bridging statistical inference and decision-making. These methodologies have been investigated across various disciplines, such as medicine, epidemiology, economics, operations research, and computer science, transcending their origins in statistics.

**Ordinal Optimization.**    Ordinal optimization involves sample allocation to each arm and selects a certain arm based on a decision-making criterion; therefore, this problem is also known as the optimal computing budget allocation problem. The development of ordinal optimization is closely related to ranking and selection problems in simulation, originating from agricultural and clinical applications in the 1950s (Gupta, 1956; Bechhofer, 1954; Paulson, 1964; Branke et al., 2007; Hong et al., 2021). A modern formulation of ordinal optimization was established in the early 2000s (Chen et al., 2000; Glynn & Juneja, 2004). Existing research has found that the probability of misidentification converges at an exponential rate for a large set of problems. By employing large deviation principles (Cramér, 1938; Ellis, 1984; Gärtner, 1977; Dembo & Zeitouni, 2009), Glynn & Juneja (2004) propose asymptotically optimal algorithms for ordinal optimization.

**BAI.**    A promising idea for enhancing the efficiency of strategies is adaptive experimental design. In this approach, information from past trials can be utilized to optimize the allocation of samples in subsequent trials. The concept of adaptive experimental design dates back to the 1970s Pong & Chow (2016). Presently, its significance is acknowledged (CDER, 2018; Chow & Chang, 2011). Adaptive strategies have also been studied within the domain of machine learning, and the multi-armed bandit (MAB) problem (Thompson, 1933; Robbins, 1952; Lai & Robbins, 1985) is an instance. The Best Arm Identification (BAI) is a paradigm of this problem (Even-Dar et al., 2006; Audibert et al., 2010; Bubeck et al., 2011), influenced by sequential testing, ranking, selection problems, and ordinal optimization (Bechhofer et al., 1968). There are two formulations in BAI: fixed-confidence (Garivier & Kaufmann, 2016) and fixed-budget BAI. In the former, the sample size (budget) is a random variable, and a decision-maker stops an experiment when a certain criterion is satisfied, similar to sequential testing Wald (1945); Chernoff (1959). In contrast, the latter fixes the sample size (budget) and minimizes a certain criterion given the sample size. BAI in this study corresponds to fixed-budget BAI (Bubeck et al., 2011; Audibert et al., 2010; Bubeck et al., 2011). There is no strict distinction between ordinal optimization and BAI.

## A.2    Optimal Strategies for BAI

We introduce arguments about optimal strategies for BAI.

**Optimal Strategies for Fixed-Confidence BAI.**    In fixed-confidence BAI, several optimal strategies have been proposed, whose expected stopping time aligns with lower bounds shown by Kaufmann et al. (2016). One of the remarkable studies is Garivier & Kaufmann (2016), which proposes an optimal strategy called Track-and-Stop by extending the Chernoff stopping rule. The Track-and-Stop strategy is refined and extended by the following studies, including Kaufmann & Koolen (2021), Jourdan et al. (2022), and Jourdan et al. (2023). From a Bayesian perspective, Russo (2020), Qin et al. (2017), and Shang et al. (2020) propose Bayesian BAI strategies that are optimal in terms of posterior convergence rate.

**Optimal Strategies for Fixed-Budget BAI.**    Fixed-budget BAI has also been extensively studied, but the asymptotic optimality has open issues. Kaufmann et al. (2016) and Carpentier & Locatelli (2016) conjecture lower bounds. Garivier & Kaufmann (2016), Kaufmann (2020), Ariu et al. (2021), and Qin (2022)

discuss and summarize the problem. In parallel with our study, Komiyama et al. (2022), Degenne (2023), Atsidakou et al. (2023), and Wang et al. (2023) further discuss the problem.

Instead of a tight evaluation of the probability of misidentification, several studies focus on evaluating the expected simple regret. Bubeck et al. (2011) discusses the optimality of uniform allocation. Kato et al. (2023a) demonstrates the asymptotic optimality of a variance-dependent strategy. Komiyama et al. (2021) develops a Bayes optimal strategy. Kato et al. (2023a) shows that variance-dependent allocation improves the expected simple regret compared to uniform allocation in Bubeck et al. (2011). However, there is a constant gap between the lower and upper bounds in Kato et al. (2023a), and it is still unknown whether allocation rules in optimal strategies depend on variances when we consider tighter evaluation in the probability of misidentification.

### A.3 Complexity of strategies and Bahadur efficiency

The complexity $-\frac{1}{T} \log \mathbb{P}_{P_0}(\widehat{a}_T^\pi \neq a^*)$, has been widely adopted in the literature of ordinal optimization and BAI (Glynn & Juneja, 2004; Kaufmann et al., 2016). In the field of hypothesis testing, Bahadur (1960) suggests the use of a similar measure to assess statistics in hypothesis testing. The efficiency of a test under the criterion proposed by Bahadur (1960) is known as Bahadur efficiency, and the complexity is referred to as the Bahadur slope. Although our problem is not hypothesis testing, it can be considered that our asymptotic optimality of strategies corresponds to the concept of Bahadur efficiency. Moreover, our global asymptotic optimality parallels global Bahadur efficiency.

### A.4 Efficient Average Treatment Effect Estimation

Efficient estimation of ATE via adaptive experiments constitutes another area of related literature. van der Laan (2008) and Hahn et al. (2011) propose experimental design methods for more efficient estimation of ATE by utilizing covariate information in treatment assignment. Despite the marginalization of covariates, their methods are able to reduce the asymptotic variance of estimators. Karlan & Wood (2014) applies the method of Hahn et al. (2011) to examine the response of donors to new information regarding the effectiveness of a charity. Subsequently, Tabord-Meehan (2022) and Kato et al. (2020) have sought to improve upon these studies, and more recently, Gupta et al. (2021) has proposed the use of instrumental variables in this context.

### A.5 Other related work.

Our problem has close ties to theories of statistical decision-making (Wald, 1949; Manski, 2000; 2002; 2004), limits of experiments (Le Cam, 1972; van der Vaart, 1998), and semiparametric theory (Hahn, 1998). The semiparametric theory is particularly crucial as it enables the characterization of lower bounds through the semiparametric analog of Fisher information (van der Vaart, 1998).

Adusumilli (2022; 2023) present a minimax evaluation of bandit strategies for both regret minimization and BAI, which is based on a formulation utilizing a diffusion process proposed by Wager & Xu (2021). Furthermore, Armstrong (2022) and Hirano & Porter (2023) extend the results of Hirano & Porter (2009) to a setting of adaptive experiments. The results of Adusumilli (2022; 2023) and Armstrong (2022) employ arguments on local asymptotic normality (Le Cam, 1960; 1972; 1986; van der Vaart, 1991; 1998), where the class of alternative hypotheses comprises "local models," in which parameters of interest converge to true parameters at a rate of $1/\sqrt{T}$

Variance-dependent strategies have garnered attention in BAI. In fixed-confidence BAI, Jourdan et al. (2023) provides a detailed discussion about BAI strategies with unknown variances. There are also studies using variances in strategies, including Sauro (2020), Lu et al. (2021), and Lalitha et al. (2023). However, they do not discuss optimality based on the arguments of Kaufmann et al. (2016). In fact, our proposed strategy differs from the one in Lalitha et al. (2023), and our results imply that at least the worst-case optimal strategy is not the one in Lalitha et al. (2023).

Extending our result, Kato et al. (2023b) discusses that we can characterize the lower bounds by using the variance when the gaps between the best and suboptimal arms approach zero (small-gap regime). This is because we can approximate the KL divergence of a wide range of distributions by the semiparametric influence function (a semiparametric analog of the Fisher information in parametric models) under the small-gap regime. Based on this finding, Kato et al. (2023b) points out that under the small-gap regime,

- when $K = 2$, and outcomes follow a distribution in the two-armed one-parameter exponential family, the uniform allocation is asymptotically optimal for consistent strategies.
- when $K \geq 3$, and outcomes follow a distribution in the two-armed one-parameter exponential family, the uniform allocation is asymptotically optimal for consistent and asymptotically invariant strategies.

This finding corresponds to a refinement of a comment by Kaufmann et al. (2016).

Furthermore, these observations are further related to Wang et al. (2023), which supports the optimality of the uniform allocation.

# B  Proof of Lemma 5.2

*Proof of Lemma 5.2.* From Lemma 5.1 and equation 2 in Definition 3.2, there exists $w^\pi$ such that

$$\limsup_{T \to \infty} -\frac{1}{T} \log \mathbb{P}_{P_0}(\widehat{a}_T^\pi \neq a^*) \leq \inf_{Q \in \cup_{b \neq a^*} \mathcal{P}(b, \underline{\Delta}, \overline{\Delta})} \sum_{a \in [K]} w^\pi(a) \mathrm{KL}(Q^a, P_0^a).$$

Then, we bound the probability as

$$\limsup_{T \to \infty} -\frac{1}{T} \log \mathbb{P}_{P_0}(\widehat{a}_T^\pi \neq a^*) \leq \sup_{w \in \mathcal{W}} \inf_{Q \in \cup_{b \neq a^*} \mathcal{P}(b, \underline{\Delta}, \overline{\Delta})} \sum_{a \in [K]} w(a) \mathrm{KL}(Q^a, P_0^a).$$

Note that $w^\pi(a)$ is independent of $Q$. Because $\mathrm{KL}(Q^a, P_0^a)$ is given as $\frac{(\mu^a(Q) - \mu^a(P_0))^2}{2(\sigma^a)^2}$ for $P, Q \in \cup_{b \neq a^*} \mathcal{P}(b, \underline{\Delta}, \overline{\Delta})$, we obtain

$$\limsup_{T \to \infty} -\frac{1}{T} \log \mathbb{P}_{P_0}(\widehat{a}_T^\pi \neq a^*) \leq \sup_{w \in \mathcal{W}} \inf_{\substack{(\mu^b) \in \mathbb{R}^K : \\ \arg\max_{b \in [K]} \mu^b \neq a^*}} \sum_{a \in [K]} w(a) \frac{(\mu^a - \mu^a(P_0))^2}{2(\sigma^a)^2}.$$

Here, we have

$$\inf_{\substack{(\mu^b) \in \mathbb{R}^K : \\ \arg\max_{b \in [K]} \mu^b \neq a^*}} \sum_{a \in [K]} w(a) \frac{(\mu^a - \mu^a(P_0))^2}{2(\sigma^a)^2}$$

$$= \min_{a \in [K] \setminus \{a^*\}} \inf_{\substack{(\mu^b) \in \mathbb{R}^K : \\ \mu^a > \mu^{a^*}}} \sum_{a \in [K]} w(a) \frac{(\mu^a - \mu^a(P_0))^2}{2(\sigma^a)^2}$$

$$= \min_{a \in [K] \setminus \{a^*\}} \inf_{\substack{(\mu^b) \in \mathbb{R}^K : \\ \mu^a > \mu^{a^*}, \ \mu^c = \mu^c(P_0)}} \sum_{a \in [K]} w(a) \frac{(\mu^a - \mu^a(P_0))^2}{2(\sigma^a)^2}$$

$$= \min_{a \in [K] \setminus \{a^*\}} \inf_{\substack{(\mu^{a^*}, \mu^a) \in \mathbb{R}^K : \\ \mu^a > \mu^{a^*}}} \left\{ w(a^*) \frac{(\mu^{a^*} - \mu^{a^*}(P_0))^2}{2(\sigma^{a^*})^2} + w(a) \frac{(\mu^a - \mu^a(P_0))^2}{2(\sigma^a)^2} \right\}$$

$$= \min_{a \in [K] \setminus \{a^*\}} \min_{\mu \in [\mu^a(P_0), \mu^{a^*}]} \left\{ w(a^*) \frac{(\mu - \mu^{a^*}(P_0))^2}{2(\sigma^{a^*})^2} + w(a) \frac{(\mu - \mu^a(P_0))^2}{2(\sigma^a)^2} \right\}.$$

Then, by solving the optimization problem, we obtain

$$\min_{a \in [K] \setminus \{a^*\}} \min_{\mu \in [\mu^a(P_0), \mu^{a^*}]} \left\{ w(a^*) \frac{(\mu - \mu^{a^*}(P_0))^2}{2(\sigma^{a^*})^2} + w(a) \frac{(\mu - \mu^a(P_0))^2}{2(\sigma^a)^2} \right\}$$

$$= \min_{a \in [K] \backslash \{a^*\}} \frac{\left(\mu^{a^*}(P_0) - \mu^a(P_0)\right)^2}{2 \left(\frac{\left(\sigma^{a^*}\right)^2}{w(a^*)} + \frac{\left(\sigma^a\right)^2}{w(a)}\right)}.$$

Thus, we complete the proof. □

## C   Proofs of Theorem 5.3

*Proof.* Because there exists $\overline{\Delta}$ such that $\mu^{a^*}(P_0) - \mu^a(P_0) \le \overline{\Delta}$ for all $a \in [K]$, the lower bound is given as

$$\min_{a \in [K] \backslash \{a^*\}} \frac{(\Delta^a(P_0))^2}{2 \left(\frac{\left(\sigma^{a^*}\right)^2}{w(a^*)} + \frac{\left(\sigma^a\right)^2}{w(a)}\right)} \le \min_{a \in [K] \backslash \{a^*\}} \frac{\overline{\Delta}^2}{2 \left(\frac{\left(\sigma^{a^*}\right)^2}{w(a^*)} + \frac{\left(\sigma^a\right)^2}{w(a)}\right)}.$$

Therefore, we consider solving

$$\max_{w \in \mathcal{W}} \min_{a \neq a^*} \frac{1}{\frac{\left(\sigma^{a^*}\right)^2}{w(a^*)} + \frac{(\sigma^a)^2}{w(a)}}.$$

We consider maximising $R > 0$ such that $R \le 1/\left\{\frac{\left(\sigma^{a^*}\right)^2}{w(a^*)} + \frac{(\sigma^a)^2}{w(a)}\right\}$ for all $a \in [K] \backslash \{a^*\}$ by optimizing $w \in \mathcal{W}$. That is, we consider the following non-linear programming:

$$\max_{R > 0, \boldsymbol{w} = \{w(1), w(2)..., w(K)\} \in (0,1)^K} R$$

$$\text{s.t.} \quad R \left(\frac{\left(\sigma^{a^*}\right)^2}{w(a^*)} + \frac{(\sigma^a)^2}{w(a)}\right) \zeta - 1 \le 0 \qquad \forall a \in [K] \backslash \{a^*\},$$

$$\sum_{a \in [K]} w(a) - 1 = 0,$$

$$w(a) > 0 \qquad \forall a \in [K].$$

The maximum of $R$ in the constraint optimization is equal to $\max_{w \in \mathcal{W}} \min_{a \neq a^*} \frac{1}{\frac{\left(\sigma^{a^*}\right)^2}{w(a^*)} + \frac{(\sigma^a)^2}{w(a)}}$.

Then, for $(K-1)$ Lagrangian multiplies $\boldsymbol{\lambda} = \{\lambda^a\}_{a \in [K] \backslash \{a^*\}}$ and $\gamma$ such that $\lambda^a \le 0$ and $\gamma \in \mathbb{R}$, we define the following Lagrangian function:

$$L(\boldsymbol{\lambda}, \boldsymbol{\gamma}; R, \boldsymbol{w}) = R + \sum_{a \in [K] \backslash \{a^*\}} \lambda^a \left\{R \left(\frac{\left(\sigma^{a^*}\right)^2}{w(a^*)} + \frac{(\sigma^a)^2}{w(a)}\right) - 1\right\} - \gamma \left\{\sum_{a \in [K]} w(a) - 1\right\}.$$

Note that the objective $(R)$ and constraints $(R \left(\frac{\left(\sigma^{a^*}\right)^2}{w(a^*)} + \frac{(\sigma^a)^2}{w(a)}\right) - 1 \le 0$ and $\sum_{a \in [K]} w(a) - 1 = 0)$ are differentiable convex functions for $R$ and $\boldsymbol{w}$. Therefore, the global optimizer $R^\dagger$ and $\boldsymbol{w}^\dagger = \{w^\dagger(a)\} \in (0, 1)^{KN}$ satisfies the KKT condition; that is, there are Lagrangian multipliers $\lambda^{a\dagger}$, $\gamma^\dagger$, and $R^\dagger$ such that

$$1 + \sum_{a \in [K] \backslash \{a^*\}} \lambda^{a\dagger} \left(\frac{\left(\sigma^{a^*}\right)^2}{w^\dagger(a^*)} + \frac{(\sigma^a)^2}{w^\dagger(a)}\right) = 0 \tag{7}$$

$$-2 \sum_{a \in [K] \backslash \{a^*\}} \lambda^{a\dagger} R^\dagger \frac{\left(\sigma^{a^*}\right)^2}{(w^\dagger(a^*))^2} = \gamma^\dagger \tag{8}$$

$$- 2\lambda^{a\dagger} R^{\dagger} \frac{(\sigma^a)^2}{(w^{\dagger}(a))^2} = \gamma^{\dagger} \qquad \forall a \in [K] \backslash \{a^*\} \tag{9}$$

$$\lambda^{a\dagger} \left\{ R^{\dagger} \left( \frac{(\sigma^{a^*})^2}{w^{\dagger}(a^*)} + \frac{(\sigma^a)^2}{w^{\dagger}(a)} \right) - 1 \right\} = 0 \qquad \forall a \in [K] \backslash \{a^*\} \tag{10}$$

$$\gamma^{\dagger} \left\{ \sum_{c \in [K]} w^{\dagger}(c) - 1 \right\} = 0$$

$$\lambda^{a\dagger} \leq 0 \qquad \forall a \in [K] \backslash \{a^*\}.$$

Here, equation 7 implies $\lambda^{a\dagger} < 0$ for some $a \in [K] \backslash \{a^*\}$. This is because if $\lambda^{a\dagger} = 0$ for all $a \in [K] \backslash \{a^*\}$, $1 + 0 = 1 \neq 0$.

With $\lambda^{a\dagger} < 0$, since $-\lambda^{a\dagger} R^{\dagger} \frac{(\sigma^a)^2}{(w^{\dagger}(a))^2} > 0$ for all $a \in [K]$, it follows that $\gamma^{\dagger} > 0$. This also implies that $\sum_{c \in [K]} w^{c\dagger} - 1 = 0$.

Then, equation 10 implies that

$$R^{\dagger} \left( \frac{(\sigma^{a^*})^2}{w^{\dagger}(a^*)} + \frac{(\sigma^a)^2}{w^{\dagger}(a)} \right) = 1 \qquad \forall a \in [K] \backslash \{a^*\}.$$

Therefore, we have

$$\frac{(\sigma^a)^2}{w^{\dagger}(a)} = \frac{(\sigma^b(P_0))^2}{w^{\dagger}(b)} \qquad \forall a, b \in [K] \backslash \{a^*\}. \tag{11}$$

Let $\frac{(\sigma^a)^2}{w^{\dagger}(a)} = \frac{(\sigma^b(P_0))^2}{w^{\dagger}(b)} = \frac{1}{R^{\dagger}} - \frac{(\sigma^{a^*})^2}{w^{\dagger}(a^*)} = U$. From equation 11 and equation 7,

$$\sum_{b \in [K] \backslash \{a^*\}} \lambda^{b\dagger} = - \frac{1}{\frac{(\sigma^{a^*})^2}{w^{\dagger}(a^*)} + U} \tag{12}$$

From equation 8 and equation 9, we have

$$\frac{(\sigma^{a^*})^2}{(w^{\dagger}(a^*))^2} \sum_{b \in [K] \backslash \{a^*\}} \lambda^{b\dagger} = \lambda^{a\dagger} \frac{(\sigma^a)^2}{(w^{\dagger}(a))^2} \qquad \forall a \in [K] \backslash \{a^*\}. \tag{13}$$

From equation 12 and equation 13, we have

$$- \frac{(\sigma^{a^*})^2}{(w^{\dagger}(a^*))^2} = \lambda^{a\dagger} \frac{(\sigma^a)^2}{(w^{\dagger}(a))^2} \left( \frac{(\sigma^{a^*})^2}{w^{\dagger}(a^*)} + U \right) \qquad \forall a \in [K] \backslash \{a^*\}. \tag{14}$$

From equation 7 and equation 14, we have

$$w^{\dagger}(a^*) = \sqrt{(\sigma^{a^*})^2 \sum_{a \in [K] \backslash \{a^*\}} \frac{(w^{\dagger}(a))^2}{(\sigma^a)^2}}.$$

In summary, we have the following KKT conditions:

$$w^{\dagger}(a^*) = \sqrt{(\sigma^{a^*})^2 \sum_{a \in [K] \backslash \{a^*\}} \frac{(w^{\dagger}(a)^2}{(\sigma^a)^2}}$$

$$\frac{\left(\sigma^{a^*}\right)^2}{(w^\dagger(a^*))^2} = -\lambda^{a\dagger}\frac{(\sigma^a)^2}{(w^\dagger(a))^2}\left(\left(\frac{\left(\sigma^{a^*}\right)^2}{w^\dagger(a^*)} + \frac{(\sigma^a)^2}{w^\dagger(a)}\right)\right) \qquad \forall a \in [K]\backslash\{a^*\}$$

$$-\lambda^{a\dagger}\frac{(\sigma^a)^2}{(w^\dagger(a))^2} = \widetilde{\gamma}^\dagger \qquad \forall a \in [K]\backslash\{a^*\}$$

$$\frac{(\sigma^a)^2}{w^\dagger(a)} = \frac{1}{R^\dagger} - \frac{\left(\sigma^{a^*}\right)^2}{w^\dagger(a^*)} \qquad \forall a \in [K]\backslash\{a^*\}$$

$$\sum_{a \in [K]} w^\dagger(a) = 1$$

$$\lambda^{a\dagger} \leq 0 \qquad \forall a \in [K]\backslash\{a^*\},$$

where $\widetilde{\gamma}^\dagger = \gamma^\dagger/2R^\dagger$. From $w^\dagger(a^*) = \sqrt{\left(\sigma^{a^*}\right)^2 \sum_{a\in[K]\backslash\{a^*\}}\frac{(w^\dagger(a))^2}{(\sigma^a)^2}}$ and $-\lambda^{a\dagger}\frac{(\sigma^a)^2}{(w^\dagger(a))^2} = \widetilde{\gamma}^\dagger$, we have

$$w^\dagger(a^*) = \sigma^{a^*}\sqrt{\sum_{a\in[K]\backslash\{a^*\}}-\lambda^{a\dagger}}/\sqrt{\widetilde{\gamma}^\dagger}$$

$$w^\dagger(a) = \sqrt{-\lambda^{a\dagger}/\widetilde{\gamma}^\dagger}\sigma^a.$$

From $\sum_{a\in[K]} w^\dagger(a) = 1$, we have

$$\sigma^{a^*}\sqrt{\sum_{a\in[K]\backslash\{a^*\}}-\lambda^{a\dagger}}/\sqrt{\widetilde{\gamma}^\dagger} + \sum_{a\in[K]\backslash\{a^*\}}\sqrt{-\lambda^{a\dagger}/\widetilde{\gamma}^\dagger}\sigma^a = 1.$$

Therefore, the following holds:

$$\sqrt{\widetilde{\gamma}^\dagger} = \sigma^{a^*}\sqrt{\sum_{a\in[K]\backslash\{a^*\}}-\lambda^{a\dagger}} + \sum_{a\in[K]\backslash\{a^*\}}\sqrt{-\lambda^{a\dagger}}\sigma^a.$$

Hence, the target allocation ratio is computed as

$$w^\dagger(a^*) = \frac{\sigma^{a^*}\sqrt{\sum_{a\in[K]\backslash\{a^*\}}-\lambda^{a\dagger}}}{\sigma^{a^*}\sqrt{\sum_{a\in[K]\backslash\{a^*\}}-\lambda^{a\dagger}} + \sum_{a\in[K]\backslash\{a^*\}}\sqrt{-\lambda^{a\dagger}}\sigma^a}$$

$$w^\dagger(a) = \frac{\sqrt{-\lambda^{a\dagger}}\sigma^a}{\sigma^{a^*}\sqrt{\sum_{a\in[K]\backslash\{a^*\}}-\lambda^{a\dagger}} + \sum_{a\in[K]\backslash\{a^*\}}\sqrt{-\lambda^{a\dagger}}\sigma^a},$$

where from $\frac{\left(\sigma^{a^*}\right)^2}{(w^\dagger(a^*))^2} = -\lambda^{a\dagger}\frac{(\sigma^a)^2}{(w^\dagger(a))^2}\left(\frac{\left(\sigma^{a^*}\right)^2}{w^\dagger(a^*)} + \frac{(\sigma^a)^2}{w^\dagger(a)}\right)$, $(\lambda^{a^*})_{a\in[K]\backslash\{a^*\}}$ satisfies,

$$\frac{1}{\sum_{a\in[K]\backslash\{a^*\}}-\lambda^{a\dagger}}$$

$$= \left(\frac{\sigma^{a^*}}{\sqrt{\sum_{a\in[K]\backslash\{a^*\}}-\lambda^{a\dagger}}} + \frac{\sigma^a}{\sqrt{-\lambda^{a\dagger}}}\right)\left(\sigma^{a^*}\sqrt{\sum_{c\in[K]\backslash\{a^*\}}-\lambda^{c\dagger}} + \sum_{c\in[K]\backslash\{a^*\}}\sqrt{-\lambda^{c\dagger}}\sigma_0^c\right)$$

$$= \left(\sigma^{a^*} + \frac{\sigma^a}{\sqrt{-\lambda^{a\dagger}}}\sqrt{\sum_{c\in[K]\backslash\{a^*\}}-\lambda^{c\dagger}}\right)\left(\sigma^{a^*} + \frac{\sum_{c\in[K]\backslash\{a^*\}}\sqrt{-\lambda^{c\dagger}}\sigma_0^c}{\sum_{c\in[K]\backslash\{a^*\}}-\lambda^{c\dagger}}\sqrt{\sum_{c\in[K]\backslash\{a^*\}}-\lambda^{c\dagger}}\right).$$

Then, the following solutions satisfy the above KKT conditions:

$$R^\dagger\left(\sigma^{a^*} + \sqrt{\sum_{b\in[K]\backslash\{a^*\}}(\sigma^b)^2}\right)^2 = 1$$

$$w^\dagger(a^*) = \frac{\sigma^{a^*}\sqrt{\sum_{b\in[K]\setminus\{a^*\}}\left(\sigma^b\right)^2}}{\sigma^{a^*}\sqrt{\sum_{b\in[K]\setminus\{a^*\}}\left(\sigma^b\right)^2} + \sum_{b\in[K]\setminus\{a^*\}}\left(\sigma^b\right)^2}$$

$$w^\dagger(a) = \frac{\left(\sigma^a\right)^2}{\sigma^{a^*}\sqrt{\sum_{b\in[K]\setminus\{a^*\}}\left(\sigma^b\right)^2} + \sum_{b\in[K]\setminus\{a^*\}}\left(\sigma^b\right)^2}$$

$$\lambda^{a\dagger} = -\left(\sigma^a\right)^2$$

$$\gamma^\dagger = 2\left(\sigma^a\right)^2.$$

$\square$

Note that a target allocation ratio $w$ in the maximum corresponds to a limit of an expectation of allocation rule $\frac{1}{T}\sum_{t=1}^{T}\mathbb{1}[A_t = a]$ from the definition of asymptotically invariant strategies.

## D  Proof of Theorem 4.1

*Proof.* Fix $P_0$ and $a^* = a^*(P_0)$. From Lemma 1 in Kaufmann et al. (2016), under any $Q \in \cup_{b\neq a^*}\mathcal{P}(b,\underline{\Delta},\overline{\Delta})$, any consistent strategy satisfies

$$\limsup_{T\to\infty} -\frac{1}{T}\log\mathbb{P}_Q(\widehat{a}_T^\pi \neq a^*) \leq \limsup_{T\to\infty}\sum_{a\in[K]}\kappa_{T,P_0}^\pi(a)\mathrm{KL}(P_0^a, Q^a).$$

Therefore, we have

$$\inf_{Q\in\cup_{b\neq a^*}\mathcal{P}(b,\underline{\Delta},\overline{\Delta})}\limsup_{T\to\infty} -\frac{1}{T}\log\mathbb{P}_Q(\widehat{a}_T^\pi \neq a^*) \leq \inf_{Q\in\cup_{b\neq a^*}\mathcal{P}(b,\underline{\Delta},\overline{\Delta})}\limsup_{T\to\infty}\sum_{a\in[K]}\kappa_{T,P_0}^\pi(a)\mathrm{KL}(P_0^a, Q^a).$$

Here, for Gaussian bandit models $P_0$ and $Q$, $\mathrm{KL}(P_0^a, Q^a) = \frac{(\mu^a(Q)-\mu^a(P_0))^2}{2(\sigma^a)^2}$ holds. Therefore,

$$\inf_{Q\in\cup_{b\neq a^*}\mathcal{P}(b,\underline{\Delta},\overline{\Delta})}\limsup_{T\to\infty} -\frac{1}{T}\log\mathbb{P}_Q(\widehat{a}_T^\pi \neq a^*(Q)) \leq \inf_{\substack{(\mu^b)\in\mathbb{R}^K:\\ \arg\max_{b\in[K]}\mu^b\neq a^*}}\limsup_{T\to\infty}\sum_{a\in[K]}\kappa_{T,P_0}^\pi(a)\frac{\left(\mu^a - \mu^a(P_0)\right)^2}{2\left(\sigma^a\right)^2},$$

which yields

$$\inf_{Q\in\cup_{b\neq a^*}\mathcal{P}(b,\underline{\Delta},\overline{\Delta})}\limsup_{T\to\infty} -\frac{1}{T}\log\mathbb{P}_Q(\widehat{a}_T^\pi \neq a^*(Q)) \leq \min_{a\in[K]\setminus\{a^*\}}\frac{\overline{\Delta}^2}{2\left(\frac{\left(\sigma^{a^*}\right)^2}{\kappa_{T,P_0}^\pi(a^*)} + \frac{\left(\sigma^a\right)^2}{\kappa_{T,P_0}^\pi(a)}\right)}$$

from the proof of Lemma 5.2 (Appendix B). Lastly, by taking $\min_{a^*\in[K]}$, we obtain

$$\min_{a^*\in[K]}\inf_{Q\in\cup_{b\neq a^*}\mathcal{P}(b,\underline{\Delta},\overline{\Delta})}\limsup_{T\to\infty} -\frac{1}{T}\log\mathbb{P}_Q(\widehat{a}_T^\pi \neq a^*(Q)) \leq \sup_{w\in\mathcal{W}}\min_{a^*\in[K]}\min_{a\in[K]\setminus\{a^*\}}\frac{\overline{\Delta}^2}{2\left(\frac{\left(\sigma^{a^*}\right)^2}{w(a^*)} + \frac{\left(\sigma^a\right)^2}{w(a)}\right)}.$$

$\square$

## E Proof of Theorem 4.2

*Proof of Theorem 4.2.* Note that the probability of misidentification can be written as

$$\mathbb{P}_{P_0}\left(\widehat{\mu}_t^{a^*} \leq \widehat{\mu}_t^{a}\right) = \mathbb{P}_{P_0}\left(\sum_{t=1}^{T}\left\{\Psi_t^{a^*}(P_0) + \Psi_t^{a}(P_0)\right\} \leq -T\Delta^a(P_0)\right),$$

where

$$\Psi_t^{a^*}(P_0) = \frac{\mathbb{1}[A_t = a^*]\left\{Y_t^{a^*} - \mu^{a^*}(P_0)\right\}}{\widetilde{w}(a^*)},$$

$$\Psi_t^{a}(P_0) = -\frac{\mathbb{1}[A_t = a]\left\{Y_t^{a} - \mu^{a}(P_0)\right\}}{\widetilde{w}(a)},$$

and $\widetilde{w}(a) = \frac{1}{T}\sum_{t=1}^{T}\mathbb{1}[A_t = a]$. Note that $\widetilde{w}(a)$ is a non-random variable by the definition of the strategy and converges to $w^{\mathrm{GNA}}(a)$ as $T \to \infty$.

By applying the Chernoff bound, for any $v < 0$ and any $\lambda < 0$, we have

$$\mathbb{P}_{P_0}\left(\sum_{t=1}^{T}\left\{\Psi_t^{a^*}(P_0) + \Psi_t^{a}(P_0)\right\} \leq v\right)$$

$$\leq \mathbb{E}_{P_0}\left[\exp\left(\lambda\sum_{t=1}^{T}\left\{\Psi_t^{a^*}(P_0) + \Psi_t^{a}(P_0)\right\}\right)\right]\exp\left(-\lambda v\right)$$

$$= \mathbb{E}_{P_0}\left[\exp\left(\lambda\sum_{t=1}^{T}\Psi_t^{a^*}(P_0)\right)\right]\mathbb{E}_{P_0}\left[\exp\left(\lambda\sum_{t=1}^{T}\Psi_t^{a}(P_0)\right)\right]\exp\left(-\lambda v\right). \tag{15}$$

First, we consider $\mathbb{E}_{P_0}\left[\exp\left(\lambda\sum_{t=1}^{T}\Psi_t^{a^*}(P_0)\right)\right]$. Because $\Psi_1^{a^*}, \Psi_2^{a^*}, \dots, \Psi_T^{a^*}$ are i.i.d., we have

$$\mathbb{E}_{P_0}\left[\exp\left(\lambda\sum_{t=1}^{T}\Psi_t^{a^*}(P_0)\right)\right] = \prod_{t=1}^{T}\mathbb{E}_{P_0}\left[\exp\left(\lambda\Psi_t^{a^*}(P_0)\right)\right] = \exp\left(\sum_{t=1}^{T}\log\mathbb{E}_{P_0}\left[\exp\left(\lambda\Psi_t^{a^*}(P_0)\right)\right]\right).$$

By applying the Taylor expansion around $\lambda = 0$, we have

$$\mathbb{E}_{P_0}\left[\exp\left(\lambda\Psi_t^{a^*}(P_0)\right)\right] = 1 + \sum_{k=1}^{\infty}\lambda^k\mathbb{E}_{P_0}\left[(\Psi_t^{a^*}(P_0))^k/k!\right].$$

holds.

Here, for $t \in [T]$ such that $A_t = a^*$,

$$\mathbb{E}_{P_0}\left[\Psi_t^{a^*}(P_0)\right] = \mathbb{E}_{P_0}\left[\frac{1}{\widetilde{w}(a^*)}\left\{Y_t^{a^*} - \mu^{a^*}(P_0)\right\}\right] = 0,$$

and

$$\mathbb{E}_{P_0}\left[\left(\Psi_t^{a^*}(P_0)\right)^2\right] = \mathbb{E}_{P_0}\left[\frac{1}{(\widetilde{w}(a^*))^2}\left\{Y_t^{a^*} - \mu^{a^*}(P_0)\right\}^2\right] = \frac{\left(\sigma^{a^*}\right)^2}{(\widetilde{w}(a^*))^2}$$

hold. Additionally, $\mathbb{E}_{P_0}\left[\left(\Psi_t^{a^*}(P_0)\right)^k\right] = 0$ holds for $k \geq 3$ because $Y^{a^*}$ follows a Gaussian distribution.

For $t \in [T]$ such that $A_t \neq a^*$, $\mathbb{E}_{P_0}\left[\left(\Psi_t^{a^*}(P_0)\right)^k\right] = 0$ hold for $k \geq 1$.

Note that the Taylor expansion of $\log(1+z)$ around $z=0$ is given as $\log(1+z) = z - z^2/2 + z^3/3 - \cdots$. Therefore, we have

$$
\begin{aligned}
&\log \mathbb{E}_{P_0}\left[\exp\left(\lambda \Psi_t^{a^*}(P_0)\right)\right] \\
&= \left\{\lambda \mathbb{E}_{P_0}\left[\Psi_t^{a^*}(P_0)\right] + \lambda^2 \mathbb{E}_{P_0}\left[(\Psi_t^{a^*}(P_0))^2/2!\right] + O\left(\lambda^3\right)\right\} - \frac{1}{2}\left\{\lambda \mathbb{E}_{P_0}\left[\Psi_t^{a^*}(P_0)\right]\right\}^2 \\
&= \lambda^2 \frac{\left(\sigma^{a^*}(P_0)\right)^2}{(\widetilde{w}(a^*))^2}.
\end{aligned}
$$

Thus,

$$
\sum_{t=1}^{T} \log \mathbb{E}_{P_0}\left[\exp\left(\lambda \Psi_t^{a^*}(P_0)\right)\right] = T\frac{\lambda^2}{2}\frac{\left(\sigma^{a^*}\right)^2}{\widetilde{w}(a^*)}
$$

holds.

Similarly,

$$
\sum_{t=1}^{T} \log \mathbb{E}_{P_0}\left[\exp\left(\lambda \Psi_t^{a}(P_0)\right)\right] = T\frac{\lambda^2}{2}\frac{\left(\sigma^{a}\right)^2}{\widetilde{w}(a)}
$$

holds.

Therefore, from equation 15, we have

$$
\mathbb{P}_{P_0}\left(\sum_{t=1}^{T}\left\{\Psi_t^{a^*}(P_0) + \Psi_t^{a}(P_0)\right\} \le v\right) \le \exp\left(T\frac{\lambda^2}{2}\frac{\left(\sigma^{a^*}\right)^2}{\widetilde{w}(a^*)} + T\frac{\lambda^2}{2}\frac{\left(\sigma^{a}\right)^2}{\widetilde{w}(a)} - \lambda v\right).
$$

Let $v = T\lambda \Omega^{a^*,a}(w^{\mathrm{GNA}})$ and $\lambda = -\frac{\Delta^a(P_0)}{\Omega^{a^*,a}(\widetilde{w})}$. Then, we have

$$
\mathbb{P}_{P_0}\left(\sum_{t=1}^{T}\left\{\Psi_t^{a^*}(P_0) + \Psi_t^{a}(P_0)\right\} \le -T\Delta^a(P_0)\right) \le \exp\left(-\frac{T\left(\Delta^a(P_0)\right)^2}{2\Omega^{a^*,a}(\widetilde{w})}\right).
$$

Finally, we have

$$
-\frac{1}{T}\log \mathbb{P}_{P_0}\left(\widehat{\mu}_T^{a^*} \le \widehat{\mu}_T^{a}\right) \ge \frac{\left(\Delta^a(P_0)\right)^2}{2\Omega^{a^*,a}(\widetilde{w})}.
$$

Because $\widetilde{w} \to w^{\mathrm{GNA}}(a)$ as $T \to \infty$, we have

$$
\liminf_{T\to\infty} -\frac{1}{T}\log \mathbb{P}_{P_0}(\widehat{a}_T^{\mathrm{EBA}} \ne a^*) \ge \liminf_{T\to\infty} -\frac{1}{T}\log \sum_{a\ne a^*} \mathbb{P}_{P_0}(\widehat{\mu}_T^{a^*} \le \widehat{\mu}_T^{a})
$$

$$
\ge \liminf_{T\to\infty} -\frac{1}{T}\log\left\{(K-1)\max_{a\ne a^*}\mathbb{P}_{P_0}(\widehat{\mu}_T^{a^*} \le \widehat{\mu}_T^{a})\right\} \ge \min_{a\ne a^*}\frac{\left(\Delta^a(P_0)\right)^2}{2\Omega^{a^*,a}(w^{\mathrm{GNA}})} \ge \min_{a\ne a^*}\frac{\Delta^2}{2\Omega^{a^*,a}(w^{\mathrm{GNA}})}.
$$

Thus, the proof of Theorem 4.2 is complete. $\qquad\square$

# F  Additional Experimental Results

This section shows the experimental results discussed in Section 7. For the details, see Section 7.

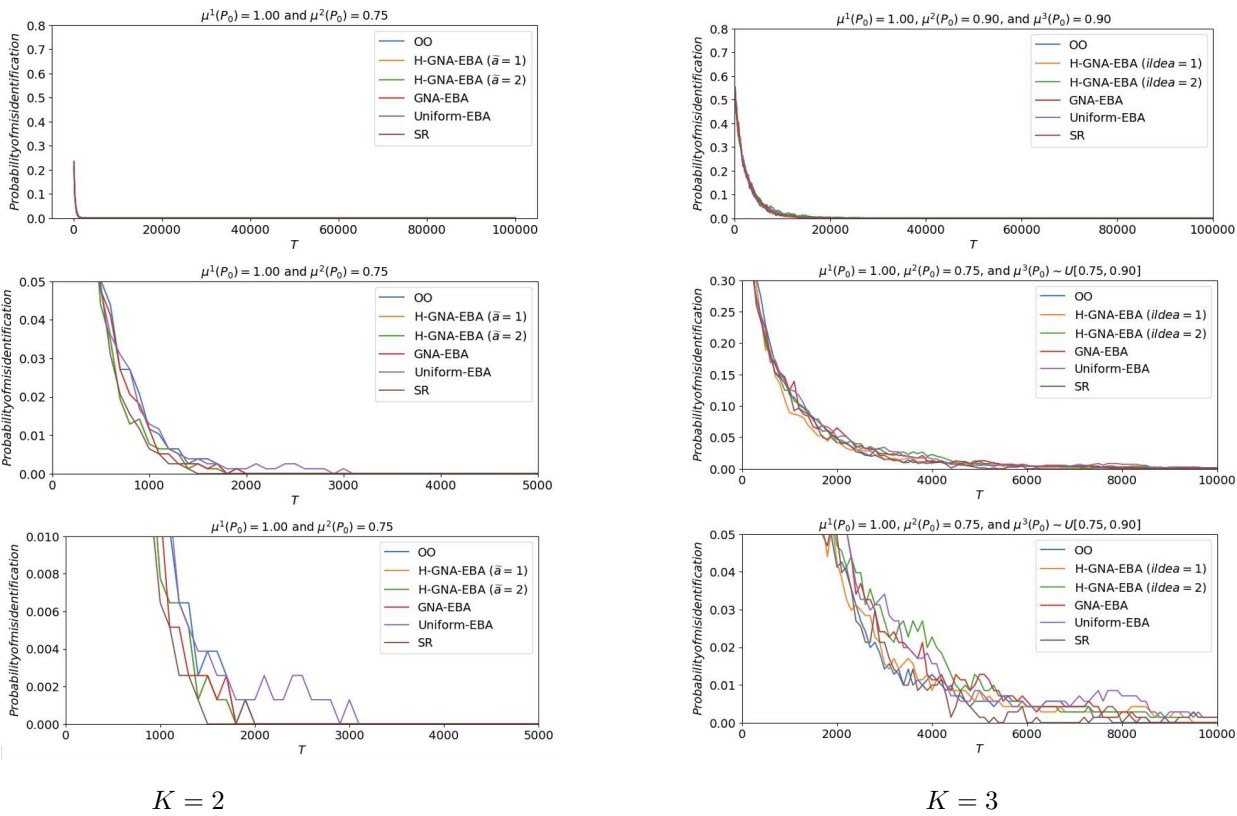

Figure 6: Experimental results ($K = 2$ and $K = 3$). The left figure shows the result with $K = 2$, and the right figure shows the result with $K = 2$. The $y$-axis and $x$-axis denote the probability of misidentification and $T$, respectively. We show the same results with different three scales of the $y$-axis and $x$-axis for each $K = 2$ and $K = 3$.

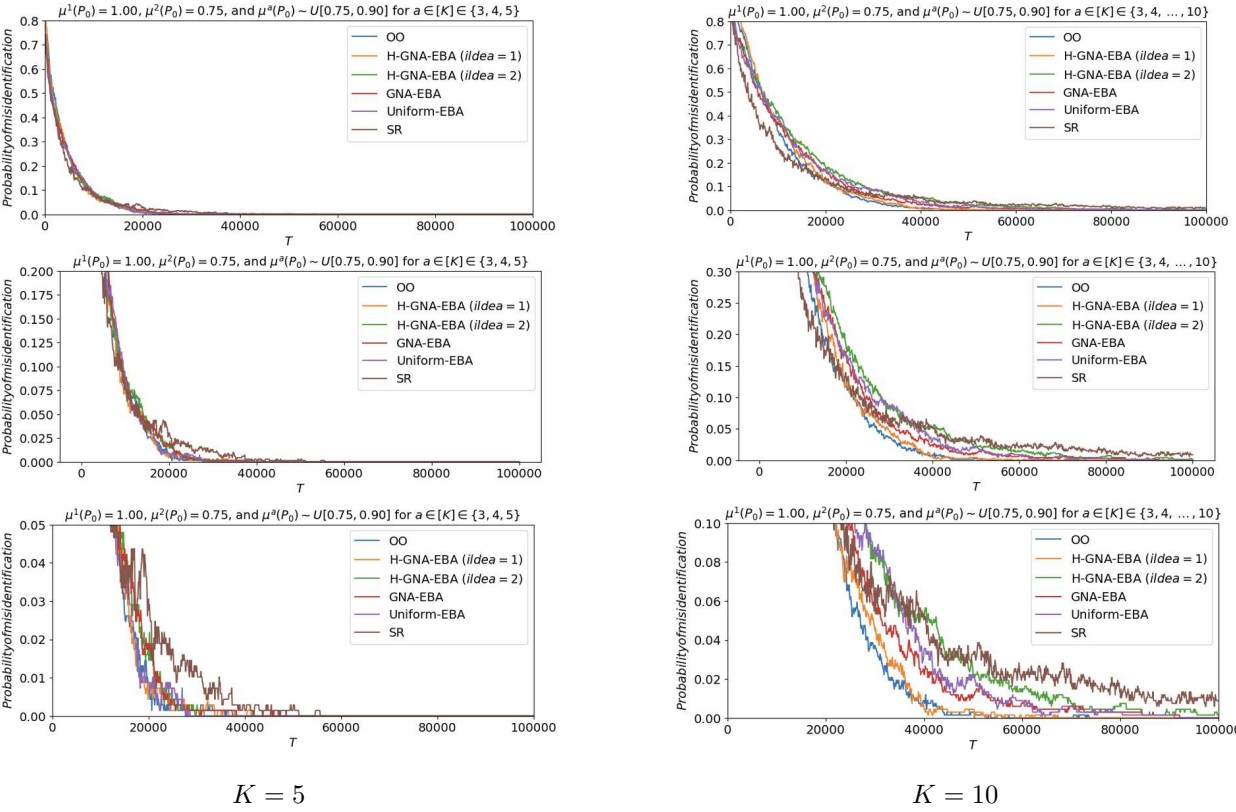

Figure 7: Experimental results ($K = 5$ and $K = 10$). The left figure shows the result with $K = 5$, and the right figure shows the result with $K = 10$. The $y$-axis and $x$-axis denote the probability of misidentification and $T$, respectively. We show the same results with different three scales of the $y$-axis and $x$-axis for each $K = 5$ and $K = 10$.

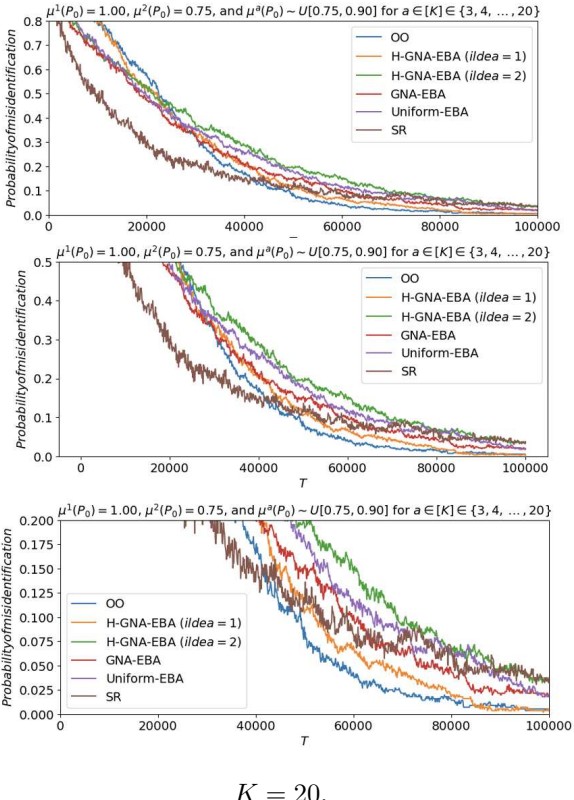

$K = 20.$

Figure 8: Experimental results ($K = 20$). The $y$-axis and $x$-axis denote the probability of misidentification and $T$, respectively. We show the same results with different three scales of the $y$-axis and $x$-axis.

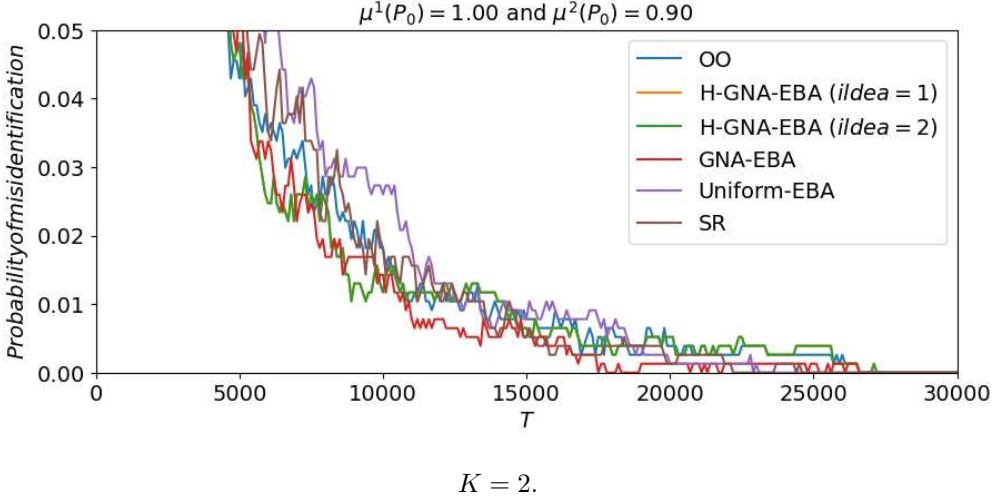

$K = 2.$

Figure 9: Experimental results. The best arm is arm 1 and $\mu^1(P) = 1$. The expected outcomes of sub-optimal arms 0.90. The variances are drawn from a uniform distribution with support $[0.5, 10]$. We continue the strategies until $T = 100,000$. We conduct 100 independent trials for each setting. For each $T \in \{100, 200, 300, \cdots, 99,900, 100,000\}$. The $y$-axis and $x$-axis denote the probability of misidentification and $T$, respectively.

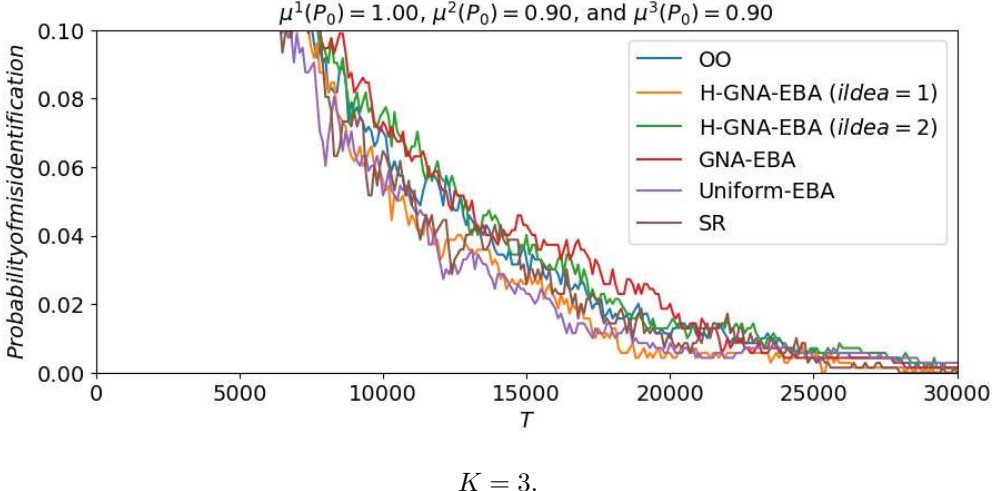

$$K = 3.$$

Figure 10: Experimental results. The best arm is arm 1 and $\mu^1(P) = 1$. The expected outcomes of suboptimal arms 0.90 ($\mu^a(P) = 0.90$ for all $a \in [K]\backslash\{1\}$). The variances are drawn from a uniform distribution with support $[0.5, 10]$. We continue the strategies until $T = 100,000$. We conduct 100 independent trials for each setting. For each $T \in \{100, 200, 300, \cdots, 99, 900, 100, 000\}$. The $y$-axis and $x$-axis denote the probability of misidentification and $T$, respectively.

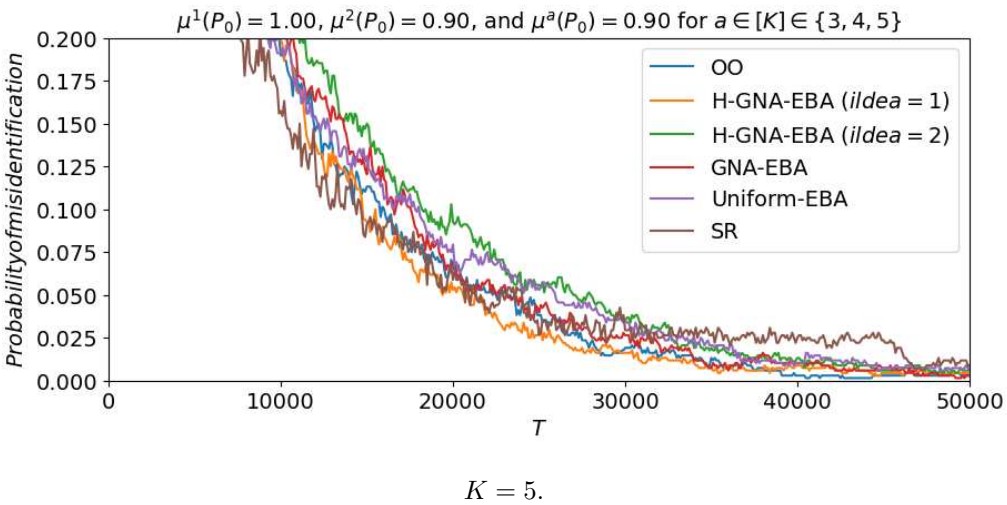

$$K = 5.$$

Figure 11: Experimental results. The best arm is arm 1 and $\mu^1(P) = 1$. The expected outcomes of suboptimal arms 0.90 ($\mu^a(P) = 0.90$ for all $a \in [K]\backslash\{1\}$). The variances are drawn from a uniform distribution with support $[0.5, 10]$. We continue the strategies until $T = 100,000$. We conduct 100 independent trials for each setting. For each $T \in \{100, 200, 300, \cdots, 99, 900, 100, 000\}$. The $y$-axis and $x$-axis denote the probability of misidentification and $T$, respectively.

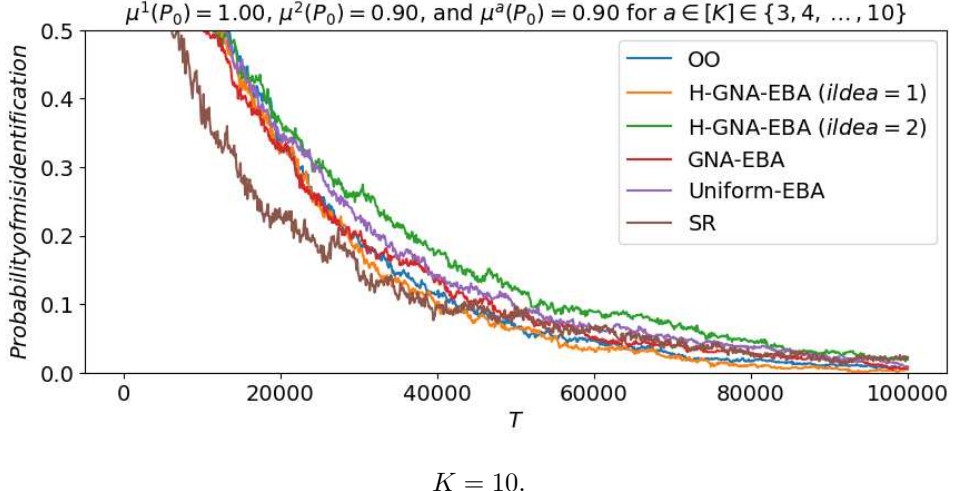

$$K = 10.$$

Figure 12: Experimental results. The best arm is arm 1 and $\mu^1(P) = 1$. The expected outcomes of suboptimal arms 0.90 ($\mu^a(P) = 0.90$ for all $a \in [K]\backslash\{1\}$). The variances are drawn from a uniform distribution with support $[0.5, 10]$. We continue the strategies until $T = 100,000$. We conduct 100 independent trials for each setting. For each $T \in \{100, 200, 300, \cdots, 99,900, 100,000\}$. The $y$-axis and $x$-axis denote the probability of misidentification and $T$, respectively.

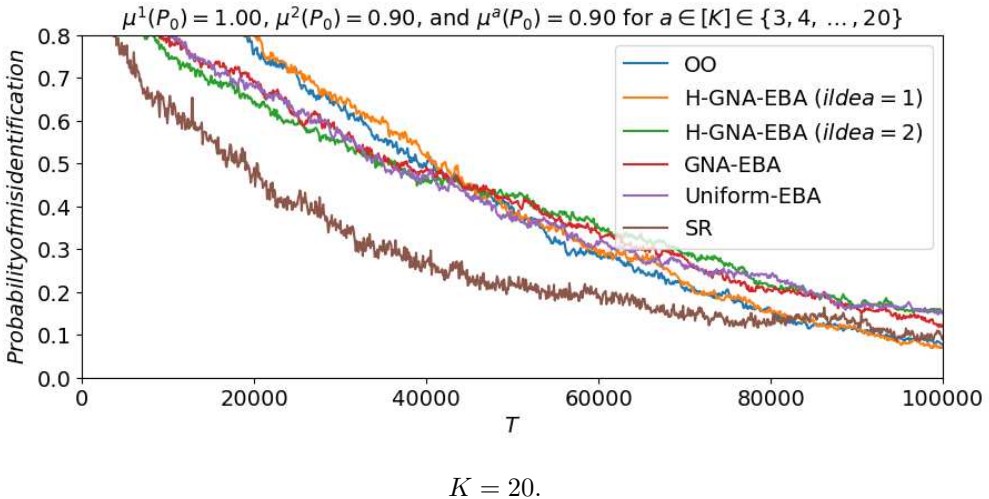

$$K = 20.$$

Figure 13: Experimental results. The best arm is arm 1 and $\mu^1(P) = 1$. The expected outcomes of suboptimal arms 0.90 ($\mu^a(P) = 0.90$ for all $a \in [K]\backslash\{1\}$). The variances are drawn from a uniform distribution with support $[0.5, 10]$. We continue the strategies until $T = 100,000$. We conduct 100 independent trials for each setting. For each $T \in \{100, 200, 300, \cdots, 99,900, 100,000\}$. The $y$-axis and $x$-axis denote the probability of misidentification and $T$, respectively.

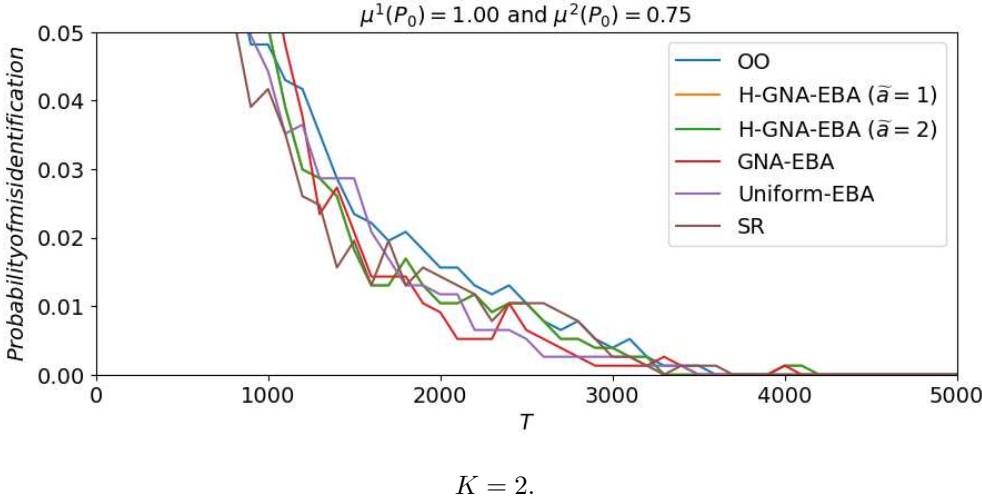

$$K = 2.$$

Figure 14: Experimental results. The best arm is arm 1 and $\mu^1(P) = 1$. The expected outcomes of suboptimal arms 0.75. The variances are drawn from a uniform distribution with support $[0.5, 10]$. We continue the strategies until $T = 100,000$. We conduct 100 independent trials for each setting. For each $T \in \{100, 200, 300, \cdots, 99,900, 100,000\}$. The $y$-axis and $x$-axis denote the probability of misidentification and $T$, respectively.

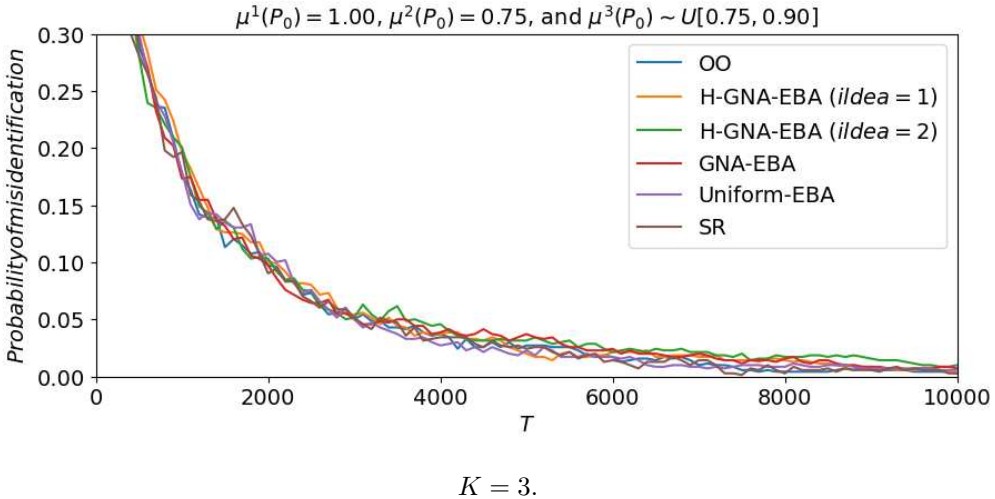

$$K = 3.$$

Figure 15: Experimental results. The expected outcomes of suboptimal arms 0.75 ($\mu^a(P) = 0.75$ for all $a \in [K] \backslash \{1\}$). The variances are drawn from a uniform distribution with support $[0.5, 10]$. We continue the strategies until $T = 100,000$. We conduct 100 independent trials for each setting. For each $T \in \{100, 200, 300, \cdots, 99,900, 100,000\}$. The $y$-axis and $x$-axis denote the probability of misidentification and $T$, respectively.

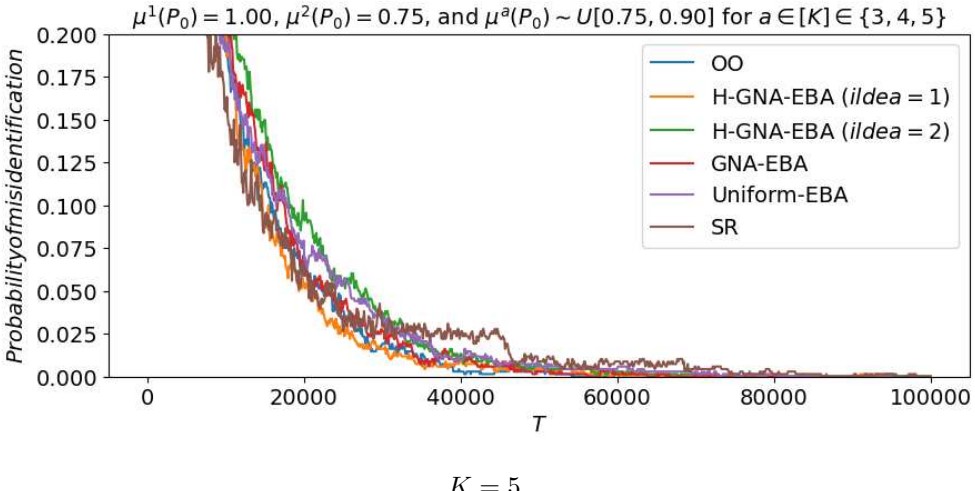

$K = 5$.

Figure 16: Experimental results. The expected outcomes of suboptimal arms 0.75 ($\mu^a(P) = 0.75$ for all $a \in [K]\backslash\{1\}$). The variances are drawn from a uniform distribution with support $[0.5, 10]$. We continue the strategies until $T = 100,000$. We conduct 100 independent trials for each setting. For each $T \in \{100, 200, 300, \cdots, 99, 900, 100, 000\}$. The $y$-axis and $x$-axis denote the probability of misidentification and $T$, respectively.

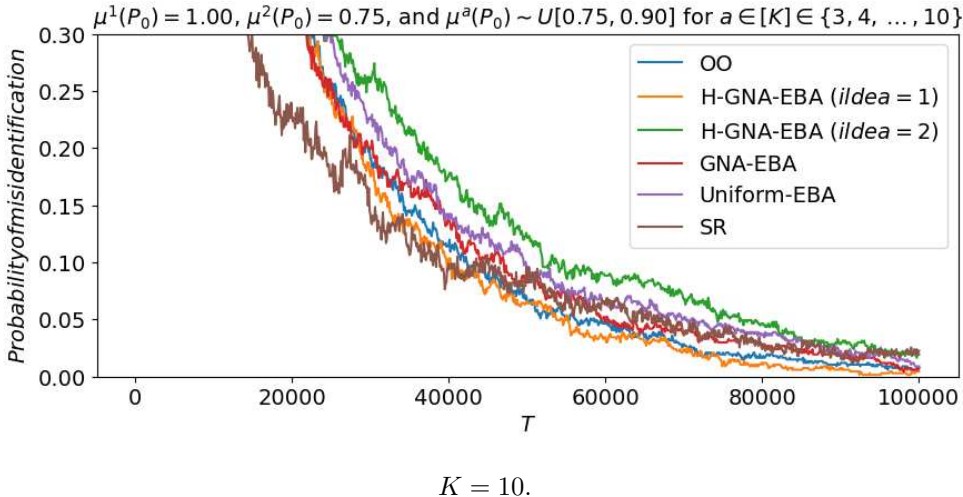

$K = 10$.

Figure 17: Experimental results. The expected outcomes of suboptimal arms 0.75 ($\mu^a(P) = 0.75$ for all $a \in [K]\backslash\{1\}$). The variances are drawn from a uniform distribution with support $[0.5, 10]$. We continue the strategies until $T = 100,000$. We conduct 100 independent trials for each setting. For each $T \in \{100, 200, 300, \cdots, 99, 900, 100, 000\}$. The $y$-axis and $x$-axis denote the probability of misidentification and $T$, respectively.

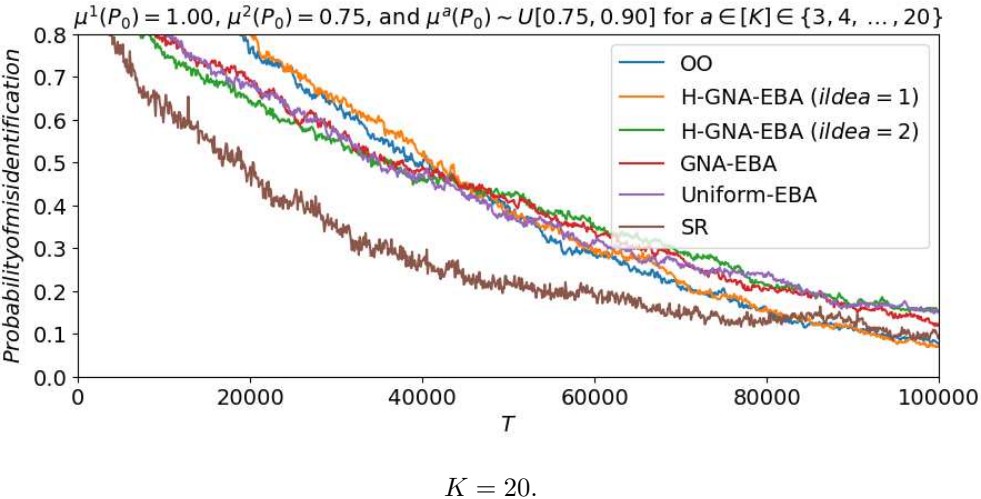

$$K = 20.$$

Figure 18: Experimental results. The expected outcomes of suboptimal arms 0.75 ($\mu^a(P) = 0.75$ for all $a \in [K]\backslash\{1\}$). The variances are drawn from a uniform distribution with support $[0.5, 10]$. We continue the strategies until $T = 100,000$. We conduct 100 independent trials for each setting. For each $T \in \{100, 200, 300, \cdots, 99,900, 100,000\}$. The $y$-axis and $x$-axis denote the probability of misidentification and $T$, respectively.

