# OpenReview forum: "Worst-Case Optimal Multi-Armed Gaussian Best Arm Identification with a Fixed Budget"
_TMLR — Rejected by TMLR_

### Review · Reviewer_XSLE · 2024-01-21

**Summary Of Contributions:**

The paper studies BAI in multi-armed Gaussian bandits with fixed budget. It provides worst-case lower bounds for the probability of misidentification. It proposes an asymptotically optimal non-adaptive strategy (GNA-EBA) whose upper bound aligns with the lower bound as the budget approaches infinity and the difference between upper and lower gaps approaches zero.

**Audience:**

No

**Broader Impact Concerns:**

not applicable.

**Claims And Evidence:**

Yes

**Requested Changes:**

1. Consider various experiment settings (changing $K, \sigma$, etc.) to diversify the simulation studies, so that the advantages and limits of the proposed algorithm is more apparent.

**Strengths And Weaknesses:**

Strengths:
1. The paper proposes first tight lower bound for this problem.
2. It proposes an algorithm which (asymmetrically) matches the lower bound.

Weaknesses:
1. The simulation studies does not clearly support the proposed algorithm. It looks like all the algorithms perform similarly. More experimental settings or similar algorithms could be added to showcase the idea.
2. Most of the bounds are not readily available (needs calculation of asymptotic expressions); this hurts the interpretation and applicability of the work. Simplifying the bounds under specific assumptions could help readability.

---

> ### Author Response · Authors · 2024-02-18
> **Reply to Reviewer XSLE**
>
> We appreciate the reviewer's constructive comments. Our responses to each question are listed below.
>
> ------
>
> ### Question 1
> > The simulation studies does not clearly support the proposed algorithm. It looks like all the algorithms perform similarly. More experimental settings or similar algorithms could be added to showcase the idea.
>
> ### Answer 1
> Thank you for your suggestion. We are currently running additional experiments and will include them as soon as they are completed. It is important to note that our study primarily aims at theoretical contributions, and thus, simulation results are not the main focus. For instance, several significant related studies, such as Degenne (2023), do not present simulation results. While we acknowledge the importance of experimental results for validation, we hope that our contributions and issues are evaluated primarily on their theoretical aspects.
>
> --------
>
> ### Question 2
> > Most of the bounds are not readily available (needs calculation of asymptotic expressions); this hurts the interpretation and applicability of the work. Simplifying the bounds under specific assumptions could help readability.
>
> ### Answer 2
> We have detailed our lower and upper bounds in newly added Sections 3 and 4, noting that our expressions are consistent with those found in existing studies. For instance, we have highlighted relevant results in Table 1. It is worth noting that, in comparison to existing works, our lower and upper bounds are relatively straightforward. For example, studies such as Glynn and Juneja (2004), Kaufmann et al. (2016), Komiyama et al. (2022), and Degenne (2023) present more complex bounds without an explicit form for the allocation ratio. In the revised draft, we have further elucidated our bounds with ample interpretations.
>
>
> Peter Glynn and Sandeep Juneja. A large deviations perspective on ordinal optimization. In Proceedings of the 2004 Winter Simulation Conference, volume 1. IEEE, 2004.
> Emilie Kaufmann, Olivier Cappé, and Aurélien Garivier. On the complexity of best-arm identification in multi-armed bandit models. Journal of Machine Learning Research, 17(1):1–42, 2016.
> Emilie Kaufmann. Contributions to the Optimal Solution of Several Bandits Problems. Habilitation á Diriger des Recherches, Université de Lille, 2020.
> Junpei Komiyama, Taira Tsuchiya, and Junya Honda. Minimax optimal algorithms for fixed-budget best arm identification. In Advances in Neural Information Processing Systems, 2022.
> Rémy Degenne. On the existence of a complexity in fixed budget bandit identification. In Conference on Learning Theory, volume 195, pp. 1131–1154. PMLR, 2023.

---

### Review · Reviewer_UfmM · 2024-01-24

**Summary Of Contributions:**

This work studied the best arm identification problem under a fixed-budget setting. It provided an asymptotic lower bound on the algorithms in Gaussian bandits, and derived au upper bound on the proposed algorithm. The bounds match in some cases. A brief set of experiments is provided.

**Audience:**

Yes

**Claims And Evidence:**

Yes

**Requested Changes:**

Please refer to the section of ''**Strengths and Weaknesses**''.

**Strengths And Weaknesses:**

**Strengths**:
1. This work provided a detailed discussion on most related works and explained what are the research gaps according to the existing literature.
2. This work assumed a multivariate Gaussian distribution on the probabilities of pulling arms when previous works such as Kaufmann et al. (2016) assumed each arm is drawn from a individual distribution. The multivariate Gaussian distribution model can be viewed as a generalization of the case where the distributions of each arm are independent.

**Weaknesses**:


This work seemed to generalize to lower bound on the error probability from the two-armed bandits to multi-armed bandits and applied the Neyman allocation techniques to provide an efficient algorithm. Overall, I'd appreciate the author(s) to highlight why such studies are non-trivial and point out the key challenges.

1. Among the first 5 pages in the 12-page main text, the author(s) focused on the literature review, which makes me doubt about the contribution of this work.
1. In Section 2.3., it states that "One of the difficulties comes from the open problem ...". May the author(s) elaborate the difficulties?
1. As the fixed-horizon BAI problem has been studied in numerous works, I suggest the author(s) to highlight what is the key contribution here.
    1. For example, as Kaufmann (2020) already provided a lower bound under the two-armed case. It that untrivial to provide a lower bound under the multiple-armed case?
1. I believe the assumptions on the distributions of arms may be different among this work and existing ones, and I'd appreciate that the author(s) can show the difference among such assumptions. Otherwise, we cannot tell whether the comparison among the results is fair.
2. This work assumed that $\underline{\Delta} \le \mu^{a^*} - \mu^a\le \bar{\Delta}$. I wonder if the consideration of $\underline{\Delta} $ and $ \bar{\Delta}$ is common. Is this just for the purpose of showing that the bounds are tight when the expected rewards of suboptimal arms are about the same?
3. Only one instance is studied in the experiment section now. I suggest the author(s) to run experiments on more instances, which will make the observation from experiments more convincing.
1. What is the $\infty$ algorithm in the experiment section?

**Minor questions**:
1. Page 2: ''Let $P_0\in \mathcal{P}(a^*, \underline{\Delta} , \bar{\Delta})$ be an instance of bandit models that generates potential outcomes in an experiment, which is decided in advance of the experiment, fixed throughout the experiment, and unknown for the decision-maker except for the variances."
     1. What does "unknown for the decision-maker except for the variances" mean?
1. Page 2: "It is known that for each fixed $P_0\in \mathcal{P}(a^*, \underline{\Delta} , \bar{\Delta})$,when $a^∗$ is unique, $P_{P_0}(\hat{a}_T=a^∗)$ converges to zero with an exponential speed as $T\rightarrow \infty$."
      1. I guess this argument comes from Definition 2.1. I suggest the author(s) to refer Definition 2.1 here.

---

> ### Author Response · Authors · 2024-02-18
> **Reply to Reviewer UfmM**
>
> We appreciate your constructive feedback. We reply to each your question below.
>
> -----
>
> ### Question 1
> > Among the first 5 pages in the 12-page main text, the author(s) focused on the literature review, which makes me doubt about the contribution of this work.
>
> ### Answer 1
> In the initial draft, we did not dedicate five pages to the literature review. The first five pages covered (i) an introduction, (ii) problem setting, (iii) open questions, and (iv) our main results, with approximately one page devoted to the literature review. To clarify, we have restructured the original Section 1 into four separate sections, Sections 1--4. In the newly added Section 3, we delve into the open questions in more detail. Section 4 now explicitly outlines our contributions.
>
> -----
>
> ### Question 2
> > In Section 2.3., it states that "One of the difficulties comes from the open problem ...". May the author(s) elaborate the difficulties?
>
> ### Answer 2
> Thank you for your suggestion. To address this, we introduced a new section, Section 3 Open Questions about Optimal Strategies in Fixed-Budget BAI, which elaborates on the open question.
>
> -----
>
> ### Question 3
> > As the fixed-horizon BAI problem has been studied in numerous works, I suggest the author(s) to highlight what is the key contribution here.
>
> ### Answer 3
> We have detailed our contributions in the newly added Section 4, particularly in Section 4.4. Our principal contribution is in establishing matching lower and upper bounds (optimal strategy) for restricted bandit models.
>
> -----
>
> ### Question 4
> **Q4.1**
> > For example, as Kaufmann (2020) already provided a lower bound under the two-armed case. It that untrivial to provide a lower bound under the multiple-armed case?
> ** Q4.2**
> I believe the assumptions on the distributions of arms may be different among this work and existing ones, and I'd appreciate that the author(s) can show the difference among such assumptions. Otherwise, we cannot tell whether the comparison among the results is fair.
>
> ### Answer 4
> Our assumptions align closely with those in Degenne (2023), with the exception that we focus on bandit models where $\overline{\Delta} = \underline{\Delta}$. Degenne (2023) demonstrated that without this model restriction, directly applying the approach of Kaufmann et al. (2016) does not result in matching bounds, a challenge identified in prior works such as Kaufmann (2020) and Ariu et al. (2021). Our work extends these findings by proving that, within the confines of restricted bandit models, matching lower and upper bounds can be achieved, drawing on the foundational work of Kaufmann et al. (2016) and hypotheses posited by Kaufmann (2020) and Garivier & Kaufmann (2016). These details are expounded in the newly added Section 3, and our findings do not contradict Degenne's conclusions.
>
> -----
>
> ### Question 5
> > This work assumed that $\underline{\Delta} \le \mu^{a^*} - \mu^a\le \bar{\Delta}$. I wonder if the consideration of  $underline{\Delta}$ and $\bar{\Delta}$  is common. Is this just for the purpose of showing that the bounds are tight when the expected rewards of suboptimal arms are about the same?
>
> ### Answer 5
> The existence of such constants (finiteness of mean parameters), although not always explicitly stated, is common in the literature, as seen in works like Kaufmann et al. (2016) and Degenne (2023). We explicitly state these assumptions to facilitate discussion on lower and upper bounds. The condition $\overline{\Delta} = \underline{\Delta}$ or $\overline{\Delta} - \underline{\Delta} \to 0$ is less typical in bandit literature but has been explored in related studies, such as Jamieson et al. (2014) and Shin et al. (2018). This regime is often considered in statistics and machine learning for parameter estimation under the concept of "localization," akin to our approach for defining optimality.
>
> Jamieson, Malloy, Nowak, and Bubeck, lil' UCB : An Optimal Exploration Algorithm for Multi-Armed Bandits, COLT 2014.
> Dongwook Shin, Mark Broadie, and Assaf Zeevi. Tractable sampling strategies for ordinal optimization. Operations Research.
>
> -----
>
> ### Question 6
> > Only one instance is studied in the experiment section now. I suggest the author(s) to run experiments on more instances, which will make the observation more convincing.
> > What is the $\infty$ algorithm in the experiment section?
>
> ### Answer 6
> Additional simulation studies are underway, and though they were not completed in time for this rebuttal, we will incorporate them as soon as possible. The symbol intended to represent the Ordinal Optimization strategy by Glynn and Juneja (2004), mistaken for $\infty$, is actually meant to denote "OO." We have clarified this in the revised draft. The OO strategy, though theoretically optimal with known true distributions, is practically infeasible. Previous research has sought a feasible lower bound to match the OO's upper bound but found it unattainable, as discussed in Ariu et al. (2021) and Degenne (2023).

---

> ### Author Response · Authors · 2024-02-18
> **Reply to Reviewer UfmM (Cont.)**
>
> ### Question 7
> > What does "unknown for the decision-maker except for the variances" mean?
>
> ### Answer 7
> The decision-maker does not known the mean parameters, including which arm is the best arm, but it knows the variances. In contrast, for example, Glynn and Juneja (2004) discusses an optimal strategy with the knowledge about the mean parameters and which arm is the best arm.
>
> ----------
>
> ### Question 8
> > I guess this argument comes from Definition 2.1. I suggest the author(s) to refer Definition 2.1 here.
>
> ### Answer 8
> The explanation comes from the uniqueness of the best arm but does not come from Definition 2.1, although it is related (Definition 2.1 is an assumption for a strategy class, while the exponential convergence is more related to the properties of the convergence theorem, such as the Markov theorem).

---

> > ### Comment · Reviewer_UfmM · 2024-02-19
> >
> > Thanks for your detailed response.

---

### Review · Reviewer_Z5wk · 2024-01-26

**Summary Of Contributions:**

This work studies the question of best-arm identification in Gaussian multi-armed bandits in the fixed-budget setting. They provide instance-dependent lower bounds on the probability of misidentifying the best arm in this setting, in particular for algorithmic strategies that are “asymptotically invariant”. They then show a minimax version of the lower bound, and propose an approach which achieves this minimax lower bound as the difference in gaps between arms go to 0.

**Audience:**

No

**Broader Impact Concerns:**

None.

**Claims And Evidence:**

Yes

**Requested Changes:**

- I believe the asymptotically invariant assumption is too strong to prove a lower bound under, and this should be relaxed (see comment above).
- Furthermore, the upper bound should be refined and shown to match the lower bound in general.
- Further discussion of and comparison to Degenne (2023) is necessary, and the discussion around whether or not there exists an optimal strategy for fixed-budget BAI should be revised in light of this to make clear that this question has been resolved (see comment above).
- Several of the results (Theorem 2.6, Theorem 4.2, and Theorem 4.3) do not have proofs. These results appear to be straightforward corollaries of other results, but some justification would still be helpful.

**Strengths And Weaknesses:**

Weaknesses:
- Asymptotically invariant strategies, which are critically used to prove the lower bounds, are defined to be strategies where asymptotically arms are played according to some fixed proportion, independent of the instance. This implies essentially that the algorithm is non-adaptive at least asymptotically—its behavior is the same on every instance. This is an extremely restrictive assumption and essentially rules out all standard bandit algorithms, which would behave differently on different instances. Therefore, I do not think lower bounds proved under this assumption are particularly interesting to those in the bandits community. (Note that while Degenne (2023) considers fixed allocations, they are allowed to vary with the instance, i.e. the lower bounds are proved with respect to the best static allocation for a given instance, not an allocation that is static across all instances—a significantly weaker assumption).
- The question of whether or not there exists an optimal strategy for fixed-budget BAI was answered by Degenne (2023). While this paper is cited, its results are barely discussed, and the discussion around whether or not there exists an optimal strategy suggests that this question is still open (which it is not).
- The upper bound assumes knowledge of the variance for each arm, which essentially makes the choice of the optimal allocation very straightforward (since the upper bound is only in the worst-case setting, the the algorithm does not need to estimate the gaps to play the proper allocation, so no learning is required to determine the correct allocation). Despite this, the upper bound only matches the lower bound as the difference between the maximum and minimum gaps for instances considered goes to 0, which is a non-standard and strong condition.
- In light of the above issue with the upper bound, it is not clear that any of the lower bounds presented in this paper, even the worst-case lower bound, are tight.

---

> ### Author Response · Authors · 2024-02-14
> **Reply to Reviewer Z5wk**
>
> We are grateful for the reviewer's constructive feedback. To begin,  **we removed the restriction of asymptotically invariant strategies by demonstrating the same lower bound only under the restriction of any consistent strategies**. Our analysis reveals that the lower bounds for any consistent strategies align with those for any asymptotically invariant strategies. Below, we respond to each comment in detail.
>
> ## Question 1
> > Asymptotically invariant strategies, which are critically used to prove the lower bounds, are defined to be strategies where asymptotically arms are played according to some fixed proportion, independent of the instance. This implies essentially that the algorithm is non-adaptive at least asymptotically—its behavior is the same on every instance. This is an extremely restrictive assumption and essentially rules out all standard bandit algorithms, which would behave differently on different instances. Therefore, I do not think lower bounds proved under this assumption are particularly interesting to those in the bandits community. (Note that while Degenne (2023) considers fixed allocations, they are allowed to vary with the instance, i.e. the lower bounds are proved with respect to the best static allocation for a given instance, not an allocation that is static across all instances—a significantly weaker assumption).
>
> ## Answer 1.
> - A1a. We would like to clarify that Degenne's fixed (static) allocation concept and our asymptotically invariant strategies are the essentially same, while ours being marginally less restrictive as it applies in the context of infinite samples. Degenne's allocation must be instance-independent regarding the data-generating process (DGP) $P_0$ to avoid the **reverse KL issue**, as Kaufmann (2020) highlights (see her page 117). An instance-dependent allocation $w$ is needed for deriving a lower bound due to its reliance on alternative hypotheses.
> - A1b. To be precise, both static allocation and our strategy may depend on a specific instance $P$. However, such an allocation $w$ must be independent of the DPG instance $P_0$. In other words, we cannot adjust the allocation depending on using the data generated from $P_0$. Otherwise, we encounter the reverse KL problem.
> - A1c. Consequently, both our strategy and static allocations could incorporate standard bandit algorithms if they prove optimal and do not depend on an instance $P_0$ that generates data (we can use an allocation depending on $P_0$ but need to fix it before observing data from $P_0$; that is, such an strategy is feasible only when we know the distribution without exploration (we need to know which arm is the best arm in advance of an experiment)). Our findings, however, indicate their suboptimality for worst-case scenarios. This conclusion is also consistent with Degenne's results.
>
> ## Question 2.
> > The question of whether or not there exists an optimal strategy for fixed-budget BAI was answered by Degenne (2023). While this paper is cited, its results are barely discussed, and the discussion around whether or not there exists an optimal strategy suggests that this question is still open (which it is not).
>
> ## Answer 2.
> - A2a. We appreciate your suggestion and have added a thorough comparison in the newly included Section 2.
> - A2b. We contend that Degenne (2023), while significantly advancing the open problem, does not close it. His contributions include:
>      -  (i) establishing lower bounds for static strategies independent of DGP instances,
>      - (ii) defining the best static strategy as the oracle, and
>      - (iii) demonstrating that lower bounds for all strategies exceed those for the oracle static strategy.
> - A2c. In contrast, our research reveals that the lower and upper bounds coincide for a specific bandit model $\mathcal{P}(\Delta)$, as defined in our new Theorem 4.4.

---

> ### Author Response · Authors · 2024-02-14
> **Reply to Reviewer Z5wk (Cont.)**
>
> ## Question 3
> > The upper bound assumes knowledge of the variance for each arm, which essentially makes the choice of the optimal allocation very straightforward (since the upper bound is only in the worst-case setting, the algorithm does not need to estimate the gaps to play the proper allocation, so no learning is required to determine the correct allocation). Despite this, the upper bound only matches the lower bound as the difference between the maximum and minimum gaps for instances considered goes to 0, which is a non-standard and strong condition.
>
> ## Answer 3
> - A3a. Although this approach might seem unconventional, it is a common practice in areas like statistical testing, where local instances are analyzed. In fixed-confidence best arm identification (BAI), Jamieson et al. (2014) focuses on a similar regime. In ordinal optimization, which is another name for fixed-budget BAI in operations research, Shin et al. (2018) discusses their strategies optimality using $\lim_{T\to\infty}-\frac{1}{T}]\log \mathbb{P}(\hat{a}\neq a^*)$ under a case where the mean gap parameter $\Delta^a(P)$ goes to zero for all $a\in[K]$.
> - A3b. Despite its restrictiveness, this framework is useful because it allows us to ignore the estimation errors of parameters in the evaluation of parameters of interest. Moreover, we filled the gap between the lower bound conjectured from Kaufmann et al. (2016) and the upper bound conjectured from Glynn and Juneja (2004) by showing the matching lower and upper bounds under the regime where we can ignore the estimation error of the allocation ratio.
> - A3c. This result is surprising since it challenges the result of Degenne (2023). Note that our result does not contradict his result.
>
> Jamieson, Malloy, Nowak, and Bubeck, lil' UCB : An Optimal Exploration Algorithm for Multi-Armed Bandits, COLT 2014.
> Dongwook Shin, Mark Broadie, and Assaf Zeevi. Tractable sampling strategies for ordinal optimization. Operations Research, 66(6):1693–1712, 2018.
>
> ## Question 4
> > Furthermore, the upper bound should be refined and shown to match the lower bound in general.
>
> ## Answer 4
> We refined the statement of the upper bound and showed that the lower and upper bounds match. We discuss this point in Theorem 4.4.
>
> ## Question 5
> > Further discussion of and comparison to Degenne (2023) is necessary, and the discussion around whether or not there exists an optimal strategy for fixed-budget BAI should be revised in light of this to make clear that this question has been resolved (see comment above).
>
> ## Answer 5.
> We added a comprehensive comparison in the newly added Section 3. Although Degenne (2023) contributes to the problem significantly, our study challenges the conclusion of Degenne (2023) and derives a different conclusion without contradicting his result.
> - Degenne (2023) shows that for any strategies, $\exists P \in \mathcal{P}^{D},\ \lim_{T\to\infty}-\frac{1}{T}\mathbb{P}(\hat{a}^\pi\neq a^*) \geq -\frac{80}{3\log(K)}\Gamma^\dagger(P),$ where $\mathcal{P}^{D}$ is Degenne (2023)'s bandit models, and $\Gamma^\dagger(P)$ is the lower bound of the oracle static allocation.
> - We show that any consistent strategies $\pi$ and our proposed strategy $\pi^{EBA}$ satisfy
> $\inf_{P\in \tilde{\mathcal{P}}}\inf_{T\to \infty}-\frac{1}{T}\log\mathbb{P}(\hat{a}^\pi \neq a^*) \leq \inf_{P\in \tilde{\mathcal{P}}}\Gamma^\dagger(P) \leq \inf_{P\in \tilde{\mathcal{P}}}\inf_{T\to \infty}-\frac{1}{T}\log\mathbb{P}(\hat{a}^{\pi^{EBA}} \neq a^*),$
> where $\tilde{\mathcal{P}}$ our bandit models where the gaps are the same across all arms (Theorem 4.4).
>
> ## Question 6.
> > Several of the results (Theorem 2.6, Theorem 4.2, and Theorem 4.3) do not have proofs. T
>
> ## Answer 6.
> We appreciate the reviewer for your suggestion. We add the proofs to them. Note that Theorem 2.6, Theomre 4.2, and Theorem 4.3 correspond to Theorem 5.4, Theorem 4.3, and Theorem 4.4.

---

> ### Author Response · Authors · 2024-02-14
> **Reply to Reviewer Z5wk (Cont.)**
>
> ## Remark
> We hope the reviewer confirms Degenne (2023), particularly the proof of Theorem 1, for his static allocation approach. If our understanding is correct, his static allocation, for example, should be stated as ``We say that a strategy is static if, under the strategy, we allocate each arm with a fixed proportion $w = (w(1), w(2),\dots, w(K))$ under any instance $P_0$ that generates the data,'' although Degenne (2023) does not provide the rigorous definition (just mentioning that "Static proportions algorithms pull all arms according to a pre-defined allocation vector in the simplex, then return the empirical correct answer."). **We again emphasize that our asymptotically invariant strategy is essentially the same as the static allocation by Degenne (2023), rather ours are slightly weaker than his. Ours and Degenne (2023)'s allocations both can depend on some $P$, but they cannot be changed for the DGP $P_0$.**
>
> Degenne (2023) shows the optimality of Glynn and Juneja (2004)'s strategy for any static allocation. However, if we only consider static allocation, we need to know which arm is the best arm, in addition to the means and variances, to implement the Glynn and Juneja (2004)'s strategy, in advance of an experiment. However, **if we know the best arm in advance, we do not have to use the BAI strategies to find the best arm (see our Section 5.5).** Our worst-case analysis reveals the static (asymptotically invariant) strategy without using such knowledge, which results in our proposed algorithm that is independent of the gap parameters. Although there may be other candidates for the static allocation that is optimal in some sense, it is still unclear.
>
> In our study, while acknowledging the significant contribution of Degenne (2023), we undertake a partial challenge to his results. Specifically, we elucidate the implications of their concept of static allocation and rigorously demonstrate that, for a certain restricted class, the lower and upper bounds coincide. In doing so, we clarify that both Degenne (2023)'s static allocation and our asymptotically invariant strategies need to predetermine the best arm without seeing data generated from the DGP instance $P_0$. Furthermore, we explore the characteristics and limitations of asymptotically invariant (static) allocations, thereby contributing to a deeper understanding of these concepts. This endeavor not only complements the existing body of knowledge but also highlights the conditions under which a theoretical lower bound is attainable in practice.
>
> If the reviewer has any questions, we will be happy to answer them.

---

### Author Response · Authors · 2024-02-14
**Reply to all reviewers.**

Thank you to all the reviewers for your constructive comments. We have revised the manuscript, with significant changes as follows:
- We have removed the restriction on asymptotically invariant strategies from the lower bound.
- Section 1 from the initial draft has been divided into the current Sections 1 to 3. Additionally, the previous Section 3 has been incorporated into the current Section 4.
- A more comprehensive literature survey has been included in the current Section 3.

Regarding the introduction to open questions, there appears to be a variety of perspectives among the reviewers. Therefore, we briefly summarize the points and reply to them below:
- RevAewer UfmM pointed out that our first 5 pages were a review of existing studies. However, these pages covered (i) an introduction, (ii) open questions, and (iii) our main results. To clarify, we have divided the original Section 1 into three separate sections.
- Reviewer UfmM suggested that a lower bound for multi-armed bandits could potentially be derived using the results from Kaufmann et al. (2016) and Kaufmann (2020). However, Kaufmann (2020) discusses the challenges of such derivation on page 117 of her work. Furthermore, as Reviewer Z5wk mentioned Degenne's result, Ariu et al. (2021) and Degenne (2023) show the non-existence of optimal strategies. Thus, deriving lower bounds from Kaufmann et al. (2016)'s results have several difficulties.
- Reviewer Z5wk pointed out that the open question was resolved by Degenne (2023). However, we believe that it remains unresolved. Degenne (2023) contributes the questions significantly but did not close it completely. Moreover, he also presents further open questions. For example, subsequent to Degenne (2023), Wang et al. (2023) explored these open questions from another perspective.
- Thus, there exists a diverse understanding of the results, as we explained above. Therefore, we expanded the literature review in Section 2 (previously Section 1.1) to clarify our contributions in the revised draft, following the Reviewer Z5wk's feedback.
- A primary open question is identifying under which conditions related to bandit models or strategy classes an upper bound strategy exists that aligns with a lower bound informed by information-theoretic analysis, as per Kaufmann et al. (2016).
- Existing studies, including works by Ariu et al. (2021) and Degenne (2023), show the existence of an instance $P\in\mathcal{P}$ under which a lower bound is larger than the conjectured lower bound based on the results from Kaufmann et al. (2016). In contrast, we examine a restricted class $\overline{\mathcal{P}} \subset \mathcal{P}$ and demonstrate that for any $P\in\overline{\mathcal{P}}$, any consistent strategies aligns with a lower bound that is closely connected to the findings of Kaufmann et al. (2016).
We believe our results are significant because, for the restricted class $\overline{\mathcal{P}}$, we have proven the existence of an asymptotically optimal strategy. Our findings offer a contrast to Degenne's conclusions but do not contradict them.

Emilie Kaufmann, Olivier Cappé, and Aurélien Garivier. On the complexity of best-arm identification in
multi-armed bandit models. Journal of Machine Learning Research, 17(1):1–42, 2016.
Emilie Kaufmann. Contributions to the Optimal Solution of Several Bandits Problems. Habilitation á Diriger des Recherches, Université de Lille, 2020.
Rémy Degenne. On the existence of a complexity in fixed budget bandit identification. In Conference on Learning Theory, volume 195, pp. 1131–1154. PMLR, 2023.
Kaito Ariu, Masahiro Kato, Junpei Komiyama, Kenichiro McAlinn, and Chao Qin. Policy choice and best
arm identification: Asymptotic analysis of exploration sampling, 2021. arXiv:2109.08229.
Po-An Wang, Kaito Ariu, and Alexandre Proutiere. On uniformly optimal algorithms for best arm identification in two-armed bandits with fixed budget, 2023. arXiv:2308.12000.

On experimental results.
- Reviewer UfmM and XSLE require more experimental studies. We are now conducting additional simulation studies. We kindly request the reviewers to grant us a little more time as our experiments are still in progress. We intend to incorporate the results into our manuscript as soon as they are available.

We will reply to the individual questions posed by the reviewers separately in our responses.

---

### Author Response · Authors · 2024-02-20
**Second Revision**

Dear all reviewers,

We revised the manuscript again. We added several additional experimental results to support our claims. Furthermore, we fix some typos in the previous drafts.

Additionally, we rewrite the definition of the asymptotically invariant strategy as
> A strategy $\pi$ is called asymptotically invariant if there exists $w^\pi \in \mathcal{W}$ such that for any DGP $P_0\in\mathcal{P}$, and all $a\in[K]$,
$$\kappa^\pi_{T, P_0}(a) = w^\pi(a) + o(1)$$
holds as $T\to \infty$. We denote the class of all possible consistent strategies by $\Pi^{\mathrm{inv}}$.

The definition is mathematically the same as the previous ones, but we clarify that the strategy is invariant against the DGP $P_0$.

Although Reviewer Z5wk points out that our asymptotically invariant strategy is restrictive compared to the static strategies by Degenne (2023), ours is (slightly) weaker than his class. If we define his static strategies mathematically, they should be defined as
> A strategy $\pi$ is called static if there exists $w^\pi \in \mathcal{W}$ such that for any DGP $P_0\in\mathcal{P}$, and all $a\in[K]$,
$$\kappa^\pi_{T, P_0}(a) = w^\pi(a)$$
holds.

As Reviewer Z5wk points out, the allocation ratio $w$ can depend on some distribution $P$. However, it cannot be constructed by using the data generated from $P_0$. This means that we can define $w$ using $P_0$, but if we define $w$ so, we cannot change it even if the data is generated from a different distribution $Q_0\neq P_0$.

Therefore, our asymptotically invariant strategy and Degenne (2023)'s static strategy both cannot include the standard strategies such as the successive halving, since they typically learn some information using data generated from $P_0$. Under the asymptotically invariant strategy, we can still learn variances or some invariant information from $P_0$ if variances or such information are fixed across $P_0\in\mathcal{P}$. However, we can make $w$ dependent on $a^*(P_0)$, which differs across $P_0\in\mathcal{P}$.

We clarified this point by revising the words in the definition.

---

### Decision · Action_Editor_tPu3 · 2024-03-02

**Recommendation:** Reject

**Comment:**

This paper proves worst-case instance-independent lower bounds for fixed-budget best-arm identification, proposes algorithms that match them, and evaluates them empirically. I like the story and see this paper accepted at TMLR. Not it in the current form though. I have two main concerns:

* Not all claims are properly supported by evidence. See "Claims And Evidence".

* The paper needs to be rewritten to be more readable and generally approachable:

  * It started as 12 pages of main text and became 20 pages after the rebuttal. Please cut it back closer to 12.

  * As one reviewer pointed out, the reader cannot wait until page 7 to get to the main contributions. Focus on what is important and leave the rest for Appendix.

I would be happy to oversee this paper again after my comments are taken into account.

**Audience:**

The paper is of a general interest to the bandit community. One thing that reduces its appeal is that GNA-EBA is not adaptive.

**Claims And Evidence:**

This paper

1. Proves worst-case instance-independent lower bounds for fixed-budget best-arm identification (BAI)

2. Proposes algorithms that match them

3. Evaluates the proposed algorithms empirically

Here is my evaluation of the evidence.

1. Great work.

2. Partial. GNA-EBA is analyzed. H-GNA-EBA is not.

3. Multiple concerns:

* The proposed algorithms use variance information. None of the baselines seems to. Please add variance-dependent baselines for Gaussian bandits, such as SHVar in Lalitha et al. (2023).

* The analyzed algorithm GNA-EBA is never the best. It seems that SR or H-GNA-EBA always outperform it.

* No error bars in the plots.

**Resubmission Of Major Revision:**

The authors may consider submitting a major revision at a later time.